# Slot-VLM: Object-Event Slots for Video-Language Modeling

**Jiaqi Xu**[1][*], **Cuiling Lan**[2], **Wenxuan Xie**[2], **Xuejin Chen**[1], **Yan Lu**[2]

[1]University of Science and Technology of China,
[2]Microsoft Research Asia

`xujiaqi@mail.ustc.edu.cn, {culan,wenxie,yanlu}@microsoft.com, xjchen99@ustc.edu.cn`

## Abstract

Video-Language Models (VLMs), powered by the advancements in Large Language Models (LLMs), are charting new frontiers in video understanding. A pivotal challenge is the development of an effective method to encapsulate video content into a set of representative tokens to align with LLMs. In this work, we introduce Slot-VLM, a new framework designed to generate semantically decomposed video tokens, in terms of object-wise and event-wise visual representations, to facilitate LLM inference. Particularly, we design an Object-Event Slots module, *i.e.*, OE-Slots, that adaptively aggregates the dense video tokens from the vision encoder to a set of representative slots. In order to take into account both the spatial object details and the varied temporal dynamics, we build OE-Slots with two branches: the Object-Slots branch and the Event-Slots branch. The Object-Slots branch focuses on extracting object-centric slots from features of high spatial resolution but low frame sample rate, emphasizing detailed object information. The Event-Slots branch is engineered to learn event-centric slots from high temporal sample rate but low spatial resolution features. These complementary slots are combined to form the vision context, serving as the input to the LLM for effective video reasoning. Our experimental results demonstrate the effectiveness of our Slot-VLM, which achieves the state-of-the-art performance on video question-answering[2].

## 1 Introduction

Recently, Large Language Models (LLMs) have gained significant progress [5, 39, 31]. They present exceptional ability to comprehend, reason with, and generate human language text. Such amazing capabilities have stimulated the wave of research on extending the models to Vision Language Models, enabling the vision reasoning ability.

For image understanding, MiniGPT-4 [51] leverages a Q-Former and a projector to align a frozen vision encoder with a frozen advanced LLM, where Q-Former converts the visual input into a fixed-length learned visual query tokens (32 tokens). LLaVA [25] and MiniGPT-v2 [10] directly use the gridded visual tokens (after projection) as the LLM input. Image-text pairs are leveraged to align the visual model and the LLM in training. For handling videos, one straightforward way is to stack the tokens from sampled frames and feed them into the LLM [19, 12, 50]. This is challenging when we densely sample frames (*e.g.*, with the purpose of preserving more information) or when we sample abundant frames for long videos. For example, for a video of 10 minutes, when we sample at 1 frame per second and each frame uses 32 (or 256), we will have 19,200 (or 153,600) tokens in total. On the one hand, the large number of tokens increases both the memory and computational requirement. On the other hand, there is spatio-temporal redundancy and the features are not representative.

---

[*]This work was done when Jiaqi was an intern at Microsoft Research Asia.
[2]This paper is the result of an open source research project starting from October, 2023.

38th Conference on Neural Information Processing Systems (NeurIPS 2024).

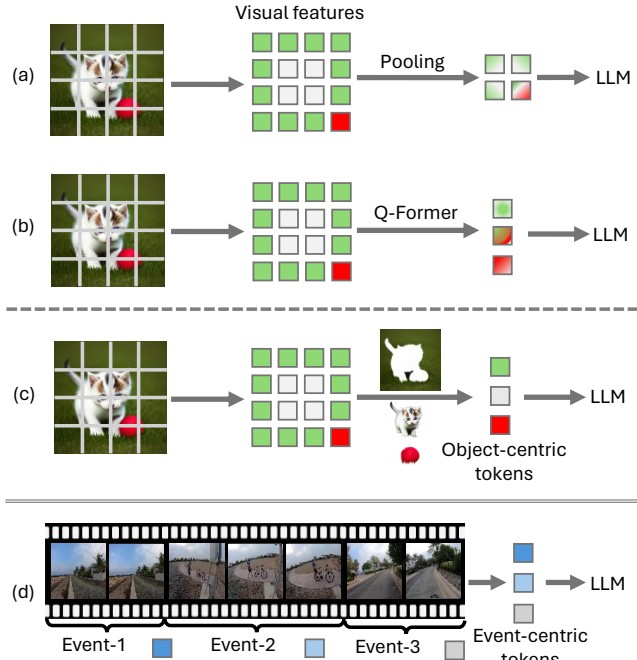

Figure 1: Illustration of methods for aligning visual features with LLM. Previous methods (a) and (b) leverage pooling or Q-Former to aggregate visual tokens, where each generated token contains *coupled* semantics. In contrast, we propose to generate semantically decoupled object-centric tokens as illustrated in (c), and event-centric tokens as illustrated in (d), to align with the LLM.

Video-ChatGPT [28] reduces the number of video tokens to 356 by separately performing spatial pooling and temporal pooling, which suffers from the loss of visual details. Video-LLaMA [48] and VideoChat [20] compress the video tokens using Q-Former, where a set of learnable queries aggregate the information from video tokens through cross-attention and self-attention. These strategies generate reduced tokens. However, each token contains semantically mixed/entangled information (see the visualization analysis of Q-Former in Appendix G.1) and this may bring inefficiency to video-language modeling. *As we know, the text words as the input to LLMs have semantics and are disentangled. Intuitively, to better align the video tokens with the language input of LLMs for building VLMs, the generated video tokens that could act similarly to words are desired.*

In this work, as illustrated in Figure 1, we aim to generate semantic-centric tokens[3] from video features to comply with LLMs for effective video-language modeling. By leveraging slot attention [27, 35], which converts an image into object-centric representations, *i.e.*, slots, we design an Object-Event Slots module (*i.e.*, OE-Slots module), that generates a set of semantically decomposed video tokens (*a.k.a.* slots) from video features, and takes such slots as input to LLMs. This is somewhat similar to human visual reasoning, where the brain converts the visual perception to high-level object representations to facilitate further reasoning. We dub the scheme powered by OE-Slots module as Slot-VLM. Figure 2 shows the flowchart of Slot-VLM. Particularly, OE-Slots module consists of two branches: Object-Slots and Event-Slots for decomposed spatial and temporal modeling, focusing on spatial objects and temporal varied dynamics (event), respectively. The Object-Slots branch extracts object-centric slots from high spatial resolution features but sampled at low frame rate. The Event-Slots branch extracts event-centric slots from high temporal resolution but low spatial resolution features. The two sets of slots are combined together as the input to the LLM for video reasoning. In instruction-tuning, we fine-tune OE-Slots module and the projection layer to align the visual features with LLM.

In summary, we have three main contributions:

---

[3]A semantic-centric token refers to a token that represents a semantically meaningful entity, such as an object, or an event (from a temporally consistent segment).

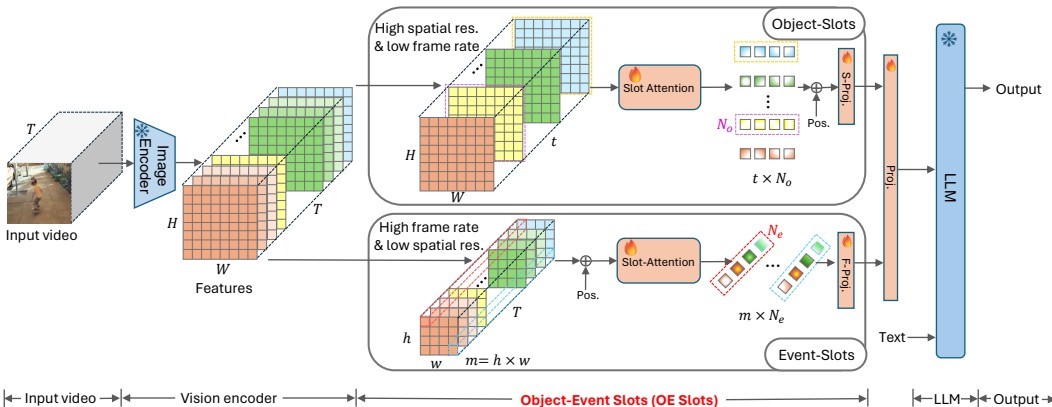

Figure 2: Flowchart of our proposed Slot-VLM for video understanding. Slot-VLM consists of a frozen image encoder, a learnable Object-Event Slots module (*i.e.*, OE-Slots module), a projection layer, and a frozen LLM. The image encoder encodes the input video of $T$ frames into a sequence of image features, resulting in extensive ($H \times W \times T$) video tokens. In order to obtain semantically decoupled and compact (reduced) video tokens as the vision context for aligning with LLM, our OE-Slots module learns to aggregate those tokens to object-centric tokens and event-centric tokens through the Object-Slots branch and the Event-Slots branch, respectively. The Object-Slots branch operates at low frame rate ($t \ll T$) but high spatial resolution in order to capture spatial objects through slot attention on each sampled frame. The Event-Slots branch operates at high frame rate but low spatial resolution ($m = h \times w$, where $h < H, w < W$) in order to capture temporal dynamics through slot attention over each spatial position. The learned slots (tokens) from two branches are projected and inputted to LLM for video reasoning, together with the text.

- We propose a new framework, Slot-VLM, for effective video-language modeling, where we generate semantically decomposed tokens to comply with LLMs. To the best of our knowledge, this work is the first to explore the use of learnable semantically decoupled visual tokens to align with LLMs. This work provides a new insight that 'disentangled' video tokens are beneficial for video-language modeling.
- We design an OE-Slots module that encourages object-centric and event-centric visual represenations for modeling spatial objects and temporal dynamics. This moves a step towards the compact and semantically meaningful token representations in VLMs.
- Experimental results demonstrate the effectiveness of our OE-Slots designs, achieving state-of-the-art performance on video question answering tasks.

We hope that this work will inspire more designs towards generating compact and semantically interpretable visual tokens for effective video-language modeling.

## 2 Related Work

**Visual Language Models** In considering the strong comprehension, reasoning, and generation ability of the LLMs, there is a surge of investigations on exploiting LLMs and vision encoders to build visual-language models, enabling the vision reasoning ability [1, 19, 51, 25, 20, 28, 49, 48, 37].

BLIP-2 [19] leverages frozen pre-trained CLIP image encoders [32] and frozen LLMs to bootstrap language-image pre-training, with the modality gap bridged by a Querying Transformer (Q-Former) and a projector. Q-Former extracts a fixed number (*e.g.*, 32) of output features (tokens) from the image feature, acting as an information bottleneck to facilitate the feeding of most useful information to LLMs. Other works like MiniGPT-4 [51] and Video-LLaMA [48] similarly leverage Q-Former to encode visual tokens, followed by image-text or video-text pair instruction tuning.

Compared with image reasoning, it is more challenging for video-language modeling, where the excessive number of visual tokens from video would enlarge the computation and memory cost to the LLM inference and thus constrains the application of VLMs. Some works sparsely sample frames and concatenate the tokens from these frames as the input to LLMs [19, 12, 50]. However, this brings

difficulty for understanding videos with high temporal dynamics or long videos, with high risk of losing valuable temporal information.

Video-ChatGPT [28] reduces the number of visual tokens by performing spatial and temporal pooling separately. LLaMA-VID [22] represents each frame with two distinct tokens: context token and content token, where context token is obtained by aggregating the visual features based on the text query while the content token is obtained by pooling the features. However, being task-agnostic, pooling would result in loss of helpful information. Moreover, the semantic-agnostic pooling inevitably results in the mixing of features of different semantics, where such coupled representations may bring difficulty for video reasoning. Q-Former provides an elegant way for generating reduced number of tokens. However, we found the learned tokens are still coupled, with abundant overlapped information among them. When aligning with the 'disentangled' word text tokens, such entanglement may result in poor compatibility with LLMs. Chat-UniVi [17] merges the dense visual tokens by clustering and uses the averaged token features in a cluster as the cluster representation. With clustering being parameter-free and task-agnostic, this is sub-optimal. It is still an open question on how to generate video tokens for constructing effective VLMs.

In this work, we explore the use of semantically decoupled tokens by learning spatial object-centric slots and temporal event-centric slots as video representation. We reveal that using semantically decoupled tokens as the input to LLMs are effective for video-language modeling.

**Object-Centric Learning** Humans naturally decompose the observed environment into entities at the appropriate level of abstraction to act in the world [35]. To obtain semantically decoupled representations, object-centric representation learning has garnered significant interest [6, 27, 36, 35]. Slot Attention [27] is a representative work. A slot attention module is designed to interface with features to produces a small set of abstract representations, named as slots, where each slot represents an object or a scene. It demonstrates encouraging object decomposition ability in simulated data but fails on real-world images [27]. To successfully learn object-centric slots in real-world images, DINOSAUR [35] learns the slots from pre-trained features by reconstructing the features. Such pre-trained features can be obtained from self-supervised learning techniques like DINO [8], MAE [15], or vision-language pre-training like CLIP [32]. These features usually have a high level of homogeneity within objects and thus facilitate the forming of object-centric representations.

In this work, we leverage slot attention to learn semantically decoupled tokens as the input to LLMs and investigate the effectiveness of aligning these 'concept tokens' with LLM input. Note that, interestingly, words in human language are abstractions of concepts, with different words tending to be disentangled. We intend to bridge the connection between vision and language by representing video with 'concepts' in terms of object-centric slots and event-centric slots.

## 3 Proposed Slot-VLM

As illustrated in Figure 2, our Slot-VLM consists of a frozen image encoder, a trainable Object-Event Slots (OE-Slots) module, a projection layer, and a frozen LLM. Given a video sequence, we extract frames at a speed of 1 frame per second (FPS) and obtain a video $V$ of $T$ frames. The video $V$ together with the text question (user instruction) is input to Slot-VLM, and Slot-VLM outputs the generated text answer.

In order to build an effective VLM for video understanding, we propose to encapsulate the dense video tokens into a small set of semantically decoupled tokens to align with the LLM. Our OE-Slots module converts the dense video tokens from the vision encoder into a set of object-centric slots (tokens) through the Object-Slots branch, and a set of event-centric slots (tokens) through the Event-Slots branch, which are then projected as the input to LLM for video reasoning.

Intriguingly, our exploitation of object-centric visual representations to enhance large language model (LLM) reasoning mirrors certain aspects of the cognitive processes humans use for visual reasoning. Human visual reasoning is a complex cognitive process. Visual signals are processed initially through the primary visual cortex (V1) and subsequently integrated in higher-level visual processing areas, resulting in the formation of complex visual object representations. Such high-level object representations together with brain-stored knowledge are then used for logical reasoning and inference to interpret observations [34, 3]. Similarly, Slot-VLM generates object-centric and event-centric token representations to provide the vision source for the LLM reasoning, where the

LLM stores rich knowledge and has strong reasoning ability, and interpretable-text generation ability. Such exploration makes us move a small step towards brain-like visual reasoning.

## 3.1 Visual Feature Extraction

To extract visual features of a video, we simply use an image-based model to get per-frame feature. Following [28], we utilize vision-language pre-trained CLIP (ViT-L/14) vision model [32] as our vision encoder. For each frame, the vision encoder outputs $H \times W$ visual tokens, with the dimension of a token denoted by $D$, where $H = W = 16$ for the CLIP vision encoder. For a sampled video of $T$ frames, we have $H \times W \times T$ visual tokens. Taking a video of 3 minutes as an example, the total number of video tokens is $16 \times 16 \times 180 = 46,080$. The large number of tokens would incur significant computational and memory costs during LLM inference. Considering the redundancy from the dense tokens, we aim to semantically exploit the the spatial structure and and temporal dynamics to reduce the number of tokens for efficient LLM inference.

## 3.2 Object-Event Slots (OE-Slots) Module

Previous methods, which aggregate the video tokens by pooling [28] or Q-Former [50, 49], actually produce tokens wherein semantics remain entangled within each token, rather than targeting at generating semantics decoupled representations. In contrast, the word tokens, which are the input to LLMs, are more semantically decoupled. In this work, we explore the using of semantics-decoupled video representations for effective video-language modeling.

We design a OE-Slots module that encapsulates the dense video tokens from the vision encoder into a small set of semantic-centric tokens to facilitate the LLM inference. It is challenging to learn semantics-centric representations from the dense video features. DINOSAUR [35] uses slot attention on the pre-trained image feature to learn object-centric representations without supervision. It is challenging to directly extend the slot learning to video features, given the substantial number of video tokens (*e.g.*, 46,080 for a 3-minute video), which increases memory and computation requirements, and the optimization difficulty.

In our work, to effectively learn semantics-centric representations from abundant video tokens, we design an Object-Event Slots module comprising two branches. The an Object-Slots branch operates at low frame rate to capture *spatial object-wise slots* for each sampled frame, while the Event-Slots branch operates at high frame rate but low spatial resolution to capture *temporal event-wise slots*. This dual-branch design is inspired by the SlowFast network [13], employing a Slow pathway for low frame rate spatial feature learning and a Fast pathway for high frame rate motion capture using a lightweight subnetwork. Unlike SlowFast which focuses on refining grid-wise features, our Object-Event Slots module aims to generate *a small set of semantically meaningful object-centric slots and event-centric slots*, serving as semantically meaningful input into LLM.

**Object-Slots Branch** As shown in Figure 2 of the Object-Slots branch, for a video of $H \times W \times T$ dense video tokens, we sample the features at low frame rate but high (original) spatial resolution to obtain $H \times W \times t$ video tokens. $t$ denotes the number of uniformly sampled frames, which we set to 8 by default.

For the $i$-th frame, we have a set $\mathcal{S}_i$ of $H \times W$ tokens. $\mathcal{S}_i$ is taken as the input to slot attention module [27, 35] to generate $N_o$ object-centric slots $\mathcal{O}_i = \{\mathbf{o}_{i,1}, \ldots, \mathbf{o}_{i,N_o}\}$ (we also refer to them as spatial slots). Please see Appendix A for more detailed formulation. Particularly, slot attention uses an iterative mechanism to map from the input tokens to the slots. At each iteration, slots attention uses cross attention with attention coefficients that are normalized over the slots (where slots are the queries) to aggregate token information. The normalization over the slot introduces competition between the slots for promoting the forming of decoupled representations.

To distinguish different frames, we add learnable temporal position embedding $\mathbf{p}_i$ to each slot of the $i$-th frame and obtain the updated object slots as $\mathcal{O}_i = \{\mathbf{o}_{i,1}, \ldots, \mathbf{o}_{i,N_o}\}$, where $\mathbf{o}_{i,j} := \mathbf{o}_{i,j} + \mathbf{p}_i$. We concatenate the updated slots of all the $t$ frames and obtain $t \times N_o$ slots $\mathcal{O} = \{\mathcal{O}_1, \ldots, \mathcal{O}_t\} = \{\mathbf{o}_{1,1}, \mathbf{o}_{1,2}, \ldots, \mathbf{o}_{1,N_o}, \ldots, \mathbf{o}_{t,N_o}\}$. A linear projection layer (S-Proj.) transforms each token to facilitate the alignment with slots from the Event-Slots branch and the alignment with the LLM.

Note that we perform spatial object slot learning on the *high spatial resolution* but *low frame rate* features in order to capture the spatial objects and reduce the number of slots, which is proportional to the number of frames.

**Event-Slots Branch** As shown in Figure 2 of the Event-Slots branch, for a video of $H \times W \times T$ dense video tokens, we sample the features at high frame rate but low spatial resolution to obtain $h \times w \times T$ video tokens. We obtain the spatial down-sampled tokens by averaging pooling with a stride of 4, therefore $h = H/4 = 4$ and $w = W/4 = 4$.

We perform temporal slot learning along temporal axis. To be aware of different frames, we add learnable temporal position embedding to each token. For the $k$-th spatial position, where $k = 1, \ldots, m$ and $m = h \times w$, we have a set $\mathcal{F}_k$ of $T$ tokens from the $T$ frames. $\mathcal{F}_k$ is taken as the input to slot attention module to generate $N_e$ event-centric slots $\mathcal{E}_k = \{\mathbf{e}_{k,1}, \ldots, \mathbf{e}_{k,N_e}\}$ (we also refer to them as temporal slots). Note that each spatial position from $h \times w$ positions corresponds to a large local patch in the pixel space, *i.e.*, 56×56 patch for a video of 224×224 spatial resolution. This allows us to observe the temporal dynamics of a large local region (56×56 pixels), which helps infer the evolution of part of an event within that region. Temporal slots aggregate the temporal tokens for explaining parts of the input, similar to identifying events. Thus, we name the learned slots as event-centric slots. We leverage temporal slots to aggregate semantically consistent temporal segments, akin to how spatial slots aggregate spatially consistent regions. While these temporal segments within a slot might not form a complete event, they are likely part of the same event.

We concatenate the slots from each of the $m$ spatial positions and obtain $m \times N_e$ slots $\mathcal{E} = \{\mathcal{E}_1, \ldots, \mathcal{E}_m\} = \{\mathbf{e}_{1,1}, \mathbf{e}_{1,2}, \ldots, \mathbf{e}_{1,N_e}, \ldots, \mathbf{e}_{m,N_e}\}$. A linear projection layer (F-Proj.) transforms each token to facilitate the alignment with slots from the Object-Slots branch and the alignment with the LLM.

Note that we perform temporal event slot learning on the *high frame rate* but *low spatial resolution* features in order to capture the temporal dynamics, and reduce the number of slots, which is proportional to the spatial feature resolution.

### 3.3 Connection to LLM

We concatenate the object-centric slots from the Object-Slots branch and the event-centric slots from the Event-Slots branch and obtain $N = t \times N_o + m \times N_e$ slots. A linear projection layer (Proj.) transforms each token to align with the LLM. The projected tokens and the text instruction are taken as the input to LLM for video reasoning.

### 3.4 Training Strategy

We divide the training procedure into three stages: slot attention pre-training, single branch instruction tuning, and two branch joint instruction tuning. We use the three stage training to better optimize the model. Stage 1 aims to pre-train the slot attention modules (with the objective of reconstructing the input features) to facilitate the learning of object-centric and event-centric slot representations, i.e., the forming of semantically decomposed tokens/slots. Stage 2 separately trains the Object-Slots branch and the Event-Slots branch to facilitate the system focusing on the optimization of each branch separately, which eases the optimization. Stage 3 jointly optimizes the two branches and the projection layer. Please see Appendix B for more details.

## 4 Experiments

### 4.1 Implementation Details

Following Video-ChatGPT [28], we employ the pre-trained CLIP vision model of ViT-L/14 [32] as the vision encoder. Note that the model size of CLIP ViT-L/14 (which we use) is 303M, which is much smaller than CLIP ViT-G/14 (1012M) as used by MovieChat [37], LLaMA-VID [22], and Video-LLaMA [23]. Specifically, we extract features from the penultimate layer, yielding an array of $H \times W$ visual tokens for each video frame. We use the Vicuna (7B) from the LLaVA model [25], to serve as our LLM. We sample at 1 fps from a video to obtain $T$ frames, and we resize each frame to 224×224 resolution. In our experiments, we set $N_o$ and $N_e$ to 8 by default unless otherwise specified. The OE-Slots module generates $N = t \times N_o + m \times N_e = 8 \times 8 + 16 \times 8 = 192$ slots (tokens)

Table 1: Comparison with the state-of-the-art methods for video QA. All these models use Vicuna-7B as the LLM. Different methods may use different datasets for pre-training. Moreover, for the instruction tuning, different methods adopt different instruction data as illustrated in the second column. For example, 11K(V)+5.1M(I) denotes the instruction data comprises about 11,000 pairs of video instructions pairs and 5.1 million pairs of image instructions. Connector denotes the method for connecting the vision features and the LLM. See Table 4 for the number of video tokens.

| Model | Instruction Data (# of Pairs) | Connector | MSVD-QA | | MSRVTT-QA | | ActivityNet-QA | | Average | |
|---|---|---|---|---|---|---|---|---|---|---|
| | | | Acc. | Score | Acc. | Score | Acc. | Score | Acc. | Score |
| Video LLaMA [48] | 11K(V)+5.1M(I) | Q-Former | 51.6 | 2.5 | 29.6 | 1.8 | 12.1 | 1.1 | 31.1 | 1.8 |
| Video Chat [20] | 11K(V)+7K(I) | Q-Former | 56.3 | 2.8 | 45 | 2.5 | 26.5 | 2.2 | 42.6 | 2.5 |
| Video-ChatGPT [28] | 100K(V) | Pooling | 64.9 | 3.3 | 49.3 | 2.8 | 35.2 | 2.7 | 49.8 | 2.9 |
| Chat-UniVi [17] | 2M(V)+433K(I) | Clustering | 65 | 3.6 | 54.6 | 3.1 | 45.8 | 3.2 | 55.1 | 3.3 |
| Video-LLaVA [23] | 100K(V) | - | 64.8 | - | 58.3 | - | 40.7 | - | 54.6 | - |
| Video-LLaVA[†] [23] | 100K(V)+665K(I) | - | 70.7 | **3.9** | 59.2 | 3.5 | 45.3 | 3.3 | 58.4 | 3.6 |
| BT-Adapter [26] | 100K(V) | Temporal Adaptor | 67 | 3.6 | 51.2 | 2.9 | 46.1 | 3.2 | 54.8 | 3.2 |
| LLaMA-VID [22] | 100K(V)+625K(I) | Q-Former&Pooling | 69.7 | 3.7 | 57.7 | 3.2 | 47.4 | 3.3 | 58.3 | 3.4 |
| VideoChat2 [21] | 0.8M(V)+1.1M(I) | Q-Former | 70 | 3.9 | 54.1 | 3.3 | 49.1 | 3.3 | 57.7 | 3.5 |
| MovieChat [37] | 11K (V)+5.1M (I) | Merge+Q-Former | 75.2 | 3.8 | 52.7 | 2.6 | 45.7 | 3.1 | 57.9 | 3.2 |
| Slot-VLM (Ours) | 100K(V) | Object-Event Slots | 74.9 | 3.8 | 69.6 | 3.4 | 48.3 | 3.4 | 64.3 | 3.5 |
| Slot-VLM[†] (Ours) | 100K(V)+665K(I) | Object-Event Slots | **75.9** | 3.8 | **69.6** | **3.5** | **49.4** | **3.4** | **65.0** | **3.6** |

from $16 \times 16 \times T$ dense video tokens. To reduce training cost, both the image encoder and LLM are frozen in our training.

All models are trained using a single NVIDIA A100 80GB GPU. The linear projection layer S-Proj., F-Proj. and Proj. consists of 1024, 1024, and 4096 neurons, respectively. We adopt a network structure that is similar to the slot attention from [36] to build the slot attention modules, with learnable slot initialization. More details please refer to Appendix B.

## 4.2 Data Details and Evaluation Metrics

We use the Video Instruction Data, collected by [28], for video instruction tuning. This comprehensive dataset comprises approximately 100K video text (question-and-answer) pairs, which are generated from the ActivityNet dataset with an average video length of 180 seconds. The dataset is characterized by a diverse array of question types. Alternatively, similar to Video-LLaVA [23], we could also incorporate a 665K image-text instruction dataset from LLaVA v1.5 [24] for enhancing the instruction tuning of the Object-Slots branch and mark the model as Slot-VLM[†]. We report both the results of Slot-VLM and Slot-VLM[†] in Table 1. Note that all our other results including ablation studies and visualization are obtained without using the 665K pairs.

We evaluate the performance on three open-ended video question-answering (QA) benchmarks like MSVD-QA[9], MSRVTT-QA[44], and ActivityNet-QA [7]. We evaluate models using accuracy (%) and average score metrics by [28], employing ChatGPT to judge prediction correctness. ChatGPT reviews each QA pair and issues a binary correctness verdict and a similarity score from 0 (least) to 5 (most) (See Appendix C for more details). We also evaluate our models on three multi-choice QA benchmarks, including Egoschema[29], NExT-QA[43] and STAR[42] (See Appendix F).

## 4.3 Comparison with the State-of-the-Arts

We evaluate the performance of our scheme against the state-of-the-art methods on three zero-shot open-ended video QA benchmarks. Table 1 shows the results. All these models use Vicuna-7B [11] as the LLM.

For video-language modeling, there is no converged standard on the training datasets, including the data for pre-training and that for instruction tuning. Generally, it is not very fair for comparisons. For example, Video-LLaMA [48] uses Webvid-2M short video dataset and CC595k image caption datasets for pre-training to enable video features to contain as much visual knowledge as possible; VideoChat2 [21] performs per-training of two stages for vision-language alignment (15M image captions from CC3M and CC12M and 10M video captions form WebVid-10M) and vision-language connection (adding 2M image captions, and 10M video captions from InternVid [40]), respectively. Similar to Video-ChatGPT [28], we do not perform such visual-text alignment pre-training. For instruction tuning, different methods use different data (see the second column in Table 1).

We have the following observations/conclusions. **1)** Our **Slot-VLM**[†] **consistently achieves the best accuracy on all the three benchmarks, outperforming all other methods, even though the volume of vision-text pairs used for our instruction tuning (0.7M pairs) is much less than many models** such as VideoChat2 [21] (1.9M pairs), Chat-UniVi [17] (2.4M pairs), and MovieChat [37] (5.1M pairs). The average performance of our Slot-VLM[†] outperforms Video-LLaVA[†] by **6.6%** in accuracy, while achieving the similar scores. **2)** In comparison to the methods highlighted in gray that utilize the same 100k video-text pairs for instruction tuning, our Slot-VLM consistently outperforms its competitors, surpassing Video-ChatGPT [28] by **10%** on MSVD-QA, **20.3%** on MSRVTT-QA, and **13.1%** on ActivityNet-QA. Furthermore, it exceeds Video-LLaVA [23] by **10.1%**, **11.3%**, and **7.6%**, and BT-Adapter [26] by **7.9%**, **18.4%**, and **2.2%** across the same benchmarks, respectively, demonstrating the high efficacy of our framework. **3)** Compared with Chat-UniVi [17] that leverages clustering to aggregate/compress tokens, our Slot-VLM[†] outperforms it significantly by **10.9%** on MSVD-QA, **15.0%** on MSRVTT-QA, and **3.6%** on ActivityNet-QA in accuracy, respectively. Note that Chat-UniVi uses 2 million video-text pairs and 433K image-text for instruction tuning while we use only 100K video-text pairs and 665K image-text pairs. **4)** Comparing Slot-VLM[†] with Slot-VLM, we can see that the incorporation of 665K image-text pairs brings an average of 0.7% gain in accuracy.

## 4.4 Ablation Studies

Table 2: Ablation studies on the effectiveness of our Slot-VLM. We compare our schemes powered by slot attention with the schemes powered by Q-Former under our framework. The FLOPs and number of parameters for the reduced token generation module are presented.

| Model | | In-domain | | MSVD-QA | | #FLOPs (G) | #Param. (M) |
|---|---|---|---|---|---|---|---|
| | | Acc. | Score | Acc. | Score | | |
| Spatial branch | Spatial-Qformer-VLM | 40.1 | 2.48 | 70.9 | 3.57 | 24.9 | 50.8 |
| | Object-Slot-VLM | **46.5** | **2.69** | **73.1** | **3.71** | 20.8 | 13.6 |
| Temporal branch | Temporal-Qformer-VLM | 46.0 | 2.64 | 72.6 | 3.62 | 26.2 | 50.8 |
| | Event-Slot-VLM | **47.1** | **2.67** | **73.1** | **3.67** | 20.7 | 13.6 |
| Two branches | ST-Qformer-VLM | 29.7 | 2.07 | 66.9 | 3.40 | 51.1 | 101.6 |
| | Slot-VLM | **48.8** | **2.75** | **74.9** | **3.76** | 41.6 | 27.3 |

We study the effectiveness of our semantics-centric designs, two branch design, and the hyper-parameter choices, on the test set of the Video Instruction Data from [28], which we refer to as In-domain, and on MSVD-QA [9]. We name our scheme with only the Object-Slots branch as Object-Slot-VLM, the scheme with only the Event-Slots branch as Event-Slot-VLM, and our final scheme as Slot-VLM.

**Effectiveness of using Semantics-centric Tokens** Under our framework, we replace our slot attention modules for generating reduced tokens by Q-Former [19], using similar training strategies and computational cost (FLOPs). Similarly, we name the schemes with the spatial branch, the temporal branch, and two branches as Spatial-QFormer-VLM, Temporal-QFormer-VLM, and ST-QFormer-VLM, respectively. Table 2 shows the results, the number of FLOPs, and the number of parameters for the reduced token generation modules, indicating the fairness for comparisons. Note we compute the FLOPs by taking a video of 100 frames. The number of parameters for our Object/Event-Slots module is much smaller than that of the Spatial/Temporal-Qformer module because the Slot Attention uses shared parameters for its $L = 3$ iterations. We can see that *the schemes using slot attention outperform those using Q-Former in both the single branch settings and the two branch setting.* Object-Slot-VLM outperforms Spatial-QFormer-VLM by 6.4% on the indomain test data (In-domain) and 2.2% on MSVD-QA, respectively. Event-Slot-VLM outperforms Temporal-QFormer-VLM by 1.1% and 0.5%. We visualized the spatial attention maps of learned Q-Former and found that Q-Former is not capable of decoupling visual tokens to semantically meaningful instances (Please see Appendix G.2).

In addition, for the two-branch scheme ST-QFormer-VLM, we found that tuning the Q-Formers and projection layers even with the single-branch trained parameters as initialization cannot lead to satisfactory results, which are poorer than the single-branch schemes. That may be caused by the difficulty in aligning the output tokens of the two branches.

**Effectiveness of our Two-branch Design** As shown in Table 2, our Slot-VLM with two branches outperforms our single-branch schemes Object-Slot-VLM by 2.3%/1.8% on In-domain/MSVD-QA

Table 3: Ablation study on the effectiveness of joint spatial-temporal slots learning vs. two branch design. The FLOPs and number of parameters for the reduced token generation module are presented.

| Model | In-domain | | MSVD-QA | | #FLOPs (G) | #Param. (M) |
|---|---|---|---|---|---|---|
| | Acc. | Score | Acc. | Score | | |
| Slot-Joint-VLM | 46.7 | 2.66 | 72.8 | 3.68 | 1304.8 | 13.7 |
| Slot-VLM | **48.8** | **2.75** | **74.9** | **3.76** | 41.6 | 27.3 |

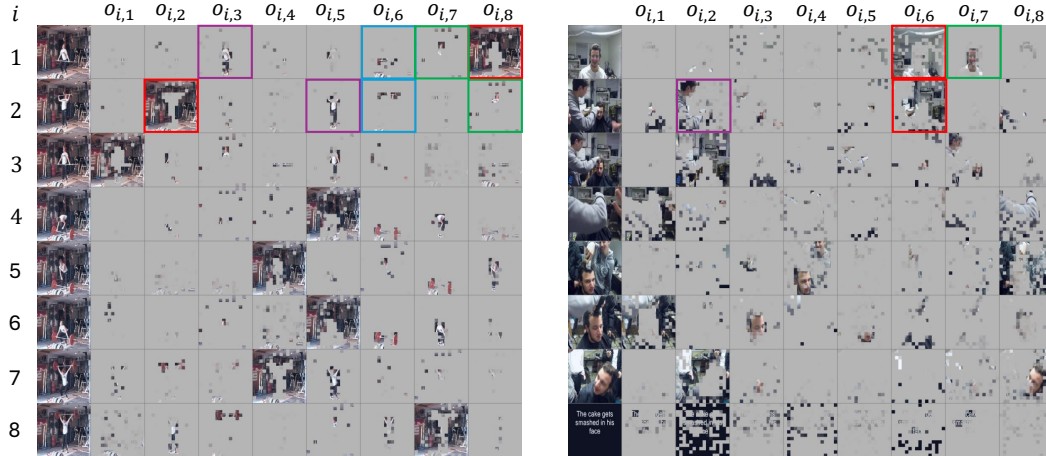

Figure 3: Visualization of spatial attention masks from the Object-Slots branch for two video examples, respectively. We have $t = 8$ frames as shown in 8 rows, indexed by $i$, where $i = 1, \ldots, t$. The first column shows the original frame. The second to the ninth columns show the cross attention mask (from slot attention) for the $N_o = 8$ object-centric slots $\mathcal{O}_i = \{\mathbf{o}_{i,1}, \ldots, \mathbf{o}_{i,N_o}\}$. We can see that even though not perfectly segmented, some meaningful slots have been formed. For example, the slots marked by red, purple, green, and blue in the first video (left) correspond to 'background', 'human body', 'head', and 'barbell'. Note that the slots in a frame is unordered and exchangeable.

and Event-Slot-VLM by 1.7%/1.8% in accuracy, respectively. Our semantics-centric tokens from the two branches are complementary.

An alternative design is to learn the same number of slots (*i.e.*, 192) from the dense $H \times W \times T$ tokens instead of from the two branched tokens. We dub the scheme as Slot-Joint-VLM. As discussed in Section 3.2, directly extending the slot learning to dense video features raises increases memory and computation requirements, and the optimization difficulty. Table 3 shows the comparison between Slot-Joint-VLM and our two-branch design Slot-VLM. We can see that the computational complexity of the reduced token generation module of Slot-VLM is much lower than that of Slot-Joint-VLM while Slot-VLM achieves superior performance.

**Influence of Hyper-parameters** Appendix E shows the influence of hyper-parameters.

### 4.5 Visualization

**Object-Centric Slots** We visualize the spatial cross-attention masks from our Slot-VLM for indicating the forming of object-centric slots for each frame, with two video samples shown in Figure 3. We can see that some meaningful slots have been formed. For example, in the first video, the slots marked by red, purple, green, and blue in the first video (left) correspond to 'background', 'human body', 'head', and 'barbell'. For comparison, we also visualize the cross-attention masks of Q-Former from the scheme Spatial-QFormer-VLM in Appendix G.2. We observe that our object-centric slots from Slot-VLM are better semantically decoupled.

Moreover, we found the instruction tuning can further promote the decoupling of slots (see Appendix G.3, where the well decoupled representations may better align with LLM.

**Event-Centric Slots** Figure 4(a) visualizes the temporal cross-attention masks from our Event-Slots branch for indicating the forming of event-centric slots for each spatial position (in total

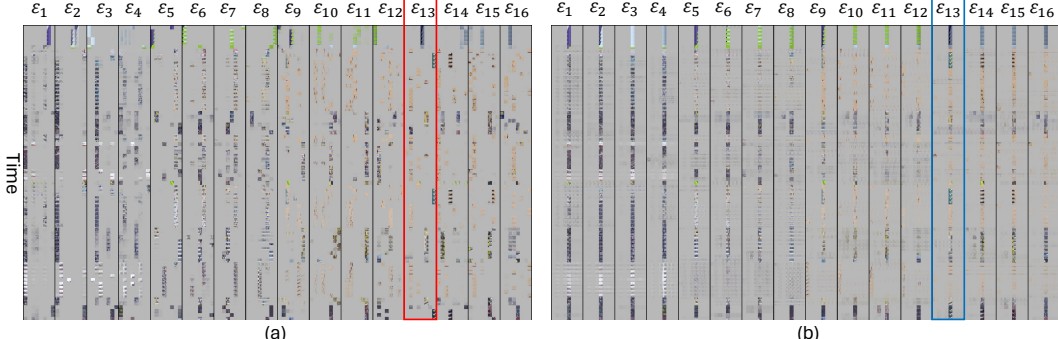

Figure 4: Visualization of temporal attention mask for $m = h \times w = 16$ spatial positions from (a) our Event-Slots branch and (b) Temporal-QFormer-VLM, respectively. For simplicity, we also refer to slot as query here. For the $k$-th spatial position, we denote the set of learned temporal queries by $\mathcal{E}_k$. Take the 13-th spatial position of the query set $\mathcal{E}_{13}$ as an example (as marked by red box in (a) and blue box in (b)). For this spatial position, the models generate $N_e = 8$ slots/queries by aggregating the temporal visual tokens. The attention masks for $\mathcal{E}_{13}$ are denoted by a map of $T$ rows and $N_e$ columns, with the visibility indicating which queries this temporal position belongs to. The higher the visibility, the greater the affinity between this temporal position and the query. We can see that in our Slot-VLM, similar contents tend to be allocated to the same slot, *i.e.*, different slots capture different contents (events) and present decoupled semantics. In contrast, in Temporal-QFormer-VLM, different contents are usually assigned to the same query or are uniformly assigned to different queries. Note that for Temporal-QFormer-VLM, we only show the mask of one head to save space, where similar observations can be found from other heads. A glimpse of the original video can be found in Figure 5. See Figure 10 for the enlarged visualization of $\mathcal{E}_{13}$.

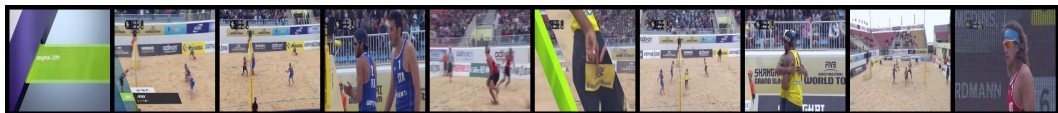

Figure 5: A glimpse of the original video used in Figure 4. For visualization purpose, we only show the frames down-sampled at a factor of 8, which is 1/8fps.

$h \times w = 16$ positions). Similarly, Figure 4(b) visualizes the temporal cross-attention masks from Temporal-QFormer-VLM for indicating the forming of query tokens for each spatial position. We can see that in our Slot-VLM, along the temporal axis, similar contents tend to be allocated to the same slot. In other words, different slots capture different contents (events) and present decoupled semantics. In contrast, in the Temporal-QFormer-VLM, different contents are usually assigned to the same query or are uniformly assigned to different queries. The queries learned from Q-Former do not present decoupled semantics. Figure 5 shows a glimpse of the original video used in Figure 4.

## 5 Conclusion

In this work, we introduce a new framework, Slot-VLM, that aims to generate a small set of semantically decoupled video tokens to comply with LLM for effective video reasoning. Particularly, we design a dual-branch Object-Event Slots module to learn object-centric slots and event-centric slots to jointly capture the spatial object details and temporal dynamics. These semantic-centric slots are taken as the input to LLM for effective video reasoning. Experimental results demonstrate the superiority of our framework and show the effectiveness of using semantics-decoupled representations for aligning with LLM. However, our current representations are still not perfect where object instances and events are not ideally segmented. We anticipate this work will inspire more investigations towards semantic-centric visual representations for video-language modeling.

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

# A More Details about Object-Slots Modeling

As shown in Figure 2 of the Object-Slots branch, for a video of $H \times W \times T$ dense video tokens, we sample the features at low frame rate but high (original) spatial resolution to obtain $H \times W \times t$ video tokens. $t$ denotes the number of uniformly sampled frames, which we set to 8 by default. In our framework, $H = W = 16$. For the $i$-th frame, we have a set $\mathcal{S}_i$ of $M = H \times W$ tokens. $\mathcal{S}_i$ is taken as the input to slot attention module [27, 35] to generate $N_o$ object-centric slots $\mathcal{O}_i = \{\mathbf{o}_{i,1}, \ldots, \mathbf{o}_{i,N_o}\}$ (we also refer to them as spatial slots).

Slot attention uses an iterative mechanism to map from the input tokens to the slots. At each iteration, slots attention uses cross attention with attention coefficients $\mathcal{A}_{m,n}$ that are first normalized over the slots

$$\mathcal{A}_{m,n} := \frac{e^{\mathcal{M}_{m,n}}}{\sum_{l=1}^{N_o} e^{\mathcal{M}_{m,l}}}, \quad \mathcal{M} := \frac{1}{D} k(\mathcal{S}_i) \cdot q(\mathcal{Q}_i)^T \in \mathbb{R}^{M \times N_o}. \tag{1}$$

Such normalization over the slots introduces competition among the slots for promoting the forming of decoupled representations. To aggregate the input tokens to their assigned slots, weighted mean is used as follows for updating slots as:

$$\widetilde{\mathcal{Q}_i} := \mathcal{W}^T \cdot v(\mathcal{S}_i) \in \mathbb{R}^{N_o \times D}, \quad \mathcal{W}_{m,n} := \frac{\mathcal{A}_{m,n}}{\sum_{l=1}^{M_o} \mathcal{A}_{l,n}}, \tag{2}$$

$k(\cdot)$, $q(\cdot)$, and $v(\cdot)$ are learnable linear transformations. Gated Recurrent Unit (GRU) and MLP parameterized by $h_\phi$ is used to further updated the slots as $\mathcal{Q}_i := h_\phi(\mathcal{Q}_i, \widetilde{\mathcal{Q}_i})$. $L$ such iterations are performed to obtain the final $N_o$ slots for this frame. Note that we initialize the slots from learnable Gaussian distributions. $L$ such iterations are performed to obtain the final $N_o$ slots for this frame. Note that we initialize the slots from learnable Gaussian distributions.

To distinguish different frames, we add learnable temporal position embedding $\mathbf{p}_i$ to each slot of the $i$-th frame and obtain the updated object slots as $\mathcal{O}_i = \{\mathbf{o}_{i,1}, \ldots, \mathbf{o}_{i,N_o}\}$, where $\mathbf{o}_{i,j} := \mathbf{o}_{i,j} + \mathbf{p}_i$. We concatenate the slots of all the $t$ frames and obtain $t \times N_o$ slots $\mathcal{O} = \{\mathcal{O}_1, \ldots, \mathcal{O}_t\} = \{\mathbf{o}_{1,1}, \mathbf{o}_{1,2}, \ldots, \mathbf{o}_{1,N_o}, \ldots, \mathbf{o}_{t,N_o}\}$. A linear projection layer (S-Proj.) transforms each token to facilitate the alignment with slots from the Event-Slots branch and the alignment with the LLM.

Note that the slot attention mechanism for learning event-centric slots is similar to that for learning object-centric slots and we will no longer elaborate on that.

# B More Implementation Details about Training

We use three stage training to better optimize the model. Stage 1 aims to pre-train the slot attention modules (with the objective of reconstructing the input features) to facilitate the learning of object-centric and event-centric slot representations, i.e., the forming of semantically decomposed tokens/slots. Stage 2 separately trains the Object-Slots branch and the Event-Slots branch to facilitate the system focusing on the optimization of each branch separately, which eases the optimization. Stage 3 jointly optimizes the two branches and the projection layer. All models are trained using a single NVIDIA A100 80GB GPU. AdamW is used as the optimizer.

**Stage 1** We first pre-train the two Slot-Attention modules of the Object-Slots branch and the Event-Slots branch separately. This promotes the formation of object-centric and event-centric slot representations. Similar to DINOSAUR [35], we use transformer decoder to reconstruct features (rather than images) obtained by CLIP ViT-L/14 by taking the outputted slots (ahead the projection) from Slot-Attention as input.

We train the models for 60 epochs with a learning rate 1e-4. We use the Slot Attention module in SLATE[36]. The dimension of each slot is set to 1024. Following [36], the number of iterations $L$ is set to 3. We use the video frames from the 100K Video Instruction Data [28] for training. Position embeddings in the Event-Slots branch are discarded during the pre-training to ease the training.

**Stage 2** We perform instruction tuning on the single branch schemes, separately. For example, we fine-tune the Object-Slots branch and the projection layers using the instruction pairs from [28], by loading the parameters from the first stage. We use the 100K Video Instruction Data from [28] to train our single branch schemes, respectively. We set the number of epochs to 3.

The Object-Slots branch/Event-Slots branch and the projection layer Proj. are tuned. We set the learning rate to 2e-5. We adopt the cosine annealing learning rate. We set the batch size to 40 and train on a single A100 GPU.

**Stage 3** We fine-tune the OE-Slots module and the projection layer using the instruction pairs, by loading the parameters from OE-Slots module from the second stage. By default, we use the Video Instruction Data for instruction tuning. We set the number of epochs to 3.

Table 4: Comparison with the state-of-the-art methods for video QA in terms of the used number of video tokens. 'Varied length' denotes the number varied with the video length. '-' denotes we did not find information from their papers. * denotes the likely number based on the paper's description

| Model | # of video tokens |
|---|---|
| Video LLaMA [48] | 256 |
| Video Chat [20] | - |
| Video-ChatGPT [28] | 356 |
| Chat-UniVi [17] | Varied length |
| Video-LLaVA [23] | - |
| Video-LLaVA† [23] | - |
| BT-Adapter [26] | 256 |
| LLaMA-VID [22] | Varied length |
| VideoChat2 [21] | 1536* |
| MovieChat [37] | - |
| Slot-VLM (Ours) | 192 |
| Slot-VLM† (Ours) | 192 |

When we incorporate 665K image-text pairs for training Slot-VLM†, we use 665K image-text pairs and 100K video-text pairs jointly to train the framework. When the input is an image-text pair, the Event-Slots branch is dropped in training. With the increased training data, we set the number of epochs to 1.

The Object-Slots branch/Event-Slots branch and the projection layer Proj. are tuned. The training hyper-parameters used in this stage are the same as those in the second stage.

Note that our three-stage training is easy to implement. Previous methods usually also use multiple stage training. For example, VideoChat2 [21] adopted a progressive three stage training using much more data than ours. LLaMA-VID [22], Video-LLaVA [23], Chat-UniVi [17] all adopted two stage training.

The numbers trainable parameters in the three stages are 27M, 37M, and 33M, respectively. The three stages training of our Slot-VLM model takes about 15, 12, and 15 hours respectively. The three stages training of our Slot-VLM† model takes about 15, 12, and 46 hours respectively.

## C  More Details about Evaluation Metrics

In our study on the open-ended video question answering, we adopt the evaluation metrics of accuracy and average score as established by [28]. To assess the accuracy of our model's predictions, we utilize ChatGPT (chatgpt35-turbo) as an evaluator. ChatGPT processes each question alongside the corresponding ground truth and the model's predicted answer. It then provides a binary "yes" or "no" judgment on the correctness of the prediction for accuracy assessment. Additionally, ChatGPT assigns an integer score ranging from 0 to 5 to quantify the closeness of the predicted answer to the ground truth, with 0 indicating no similarity and 5 denoting a close match.

## D  Number of Video Tokens Used by Different Models

Table 4 shows the performance gain is not due to the final number of video tokens input to the LLMs, where our Slot-VLM uses only 192 tokens that is much smaller than some other models.

## E  More Ablation Studies

**Influence of High Spatial Resolution for the Object-Slots Branch and High Frame Rate for the Event-Slots Branch** We study the influence of spatial resolution of the visual features in the Object-Slots branch and the influence of frame rate of the visual features in the Event-Slots branch, respectively. As shown in Table 5, when we reduce the spatial resolution from $16 \times 16$ to $4 \times 4$ for the Object-Slots branch, the performance drops by 4.6%/3.3% on In-domain/MSVD-QA in accuracy. When we reduce the frame rate by a factor of 8 from 1fps to 1/8fps for the Event-Slots branch, the performance drops by 4.3%/1.4% on In-domain/MSVD-QA in accuracy.

**Influence of Hyperparameter $N_o$** We study the influence of the number of object-centric slots $N_o$ of a frame for Object-Slot-VLM and show the results in Table 6. We can see that as the increase of the number of slots from 4 to 16 for a frame, the performance increases and saturates when $N_o = 8$ in terms of score in In-domain. Therefore, by default, we set $N_o = 8$.

Table 5: Ablation study on the influence of high spatial resolution for the Object-Slots branch and high frame rate for the Event-Slots branch. Object-Slot-VLM ($4 \times 4$) denotes the spatial resolution is reduced from $16 \times 16$ to $4 \times 4$. Event-Slot-VLM ($T/8$) denotes the frame rate is reduced by a factor of 8.

| Model | | In-domain | | MSVD-QA | |
|---|---|---|---|---|---|
| | | Acc. | Score | Acc. | Score |
| Object-Slots branch | Object-Slot-VLM (4x4) | 41.9 | 2.53 | 69.8 | 3.55 |
| | Object-Slot-VLM | **46.5** | **2.69** | **73.1** | **3.71** |
| Event-Slots branch | Event-Slot-VLM (T/8) | 42.9 | 2.59 | 71.6 | 3.63 |
| | Event-Slot-VLM | **47.1** | **2.67** | **73.1** | **3.67** |

Table 6: Ablation study on the influence of the number of object-centric slots $N_o$ and the influence of the number of event-centric slots $N_e$, respectively.

| Model | | In-domain | | MSVD-QA | |
|---|---|---|---|---|---|
| | | Acc. | Score | Acc. | Score |
| Object-Slots branch | Object-Slot-VLM (4slots) | 42.6 | 2.54 | 72.7 | 3.67 |
| | Object-Slot-VLM (8slots) | 46.5 | **2.69** | **73.1** | **3.71** |
| | Object-Slot-VLM (16slots) | **47.7** | **2.69** | 72.9 | 3.70 |
| Event-Slots branch | Event-Slot-VLM (4slots) | 46.4 | 2.66 | 74.2 | **3.69** |
| | Event-Slot-VLM (8slots) | **47.1** | 2.67 | 73.1 | 3.67 |
| | Event-Slot-VLM (16slots) | 46.3 | **2.68** | **74.4** | **3.69** |

**Influence of Hyperparameter** $N_e$ We study the influence of the number of event-centric slots $N_e$ of a spatial position for Event-Slot-VLM and show the results in Table 6. We can see that as the increase of the number of slots from 4 to 16 along the temporal direction, the performance increases in terms of score in In-domain and achieves good performance when $N_e = 8$. For simplicity, we set $N_e = 8$ by default.

**Influence of Frame Sampling Rate in the Object-Slots Branch and Stride for Pooling in the Event-Slots Branch** We conducted ablation studies using single branch settings. For the Object-Slots branch, we tested three frame sampling rates: 4 frames (Object-Slot-VLM (4 frames)), 8 frames, and 16 frames per video. For the Event-Slots branch, we tested four pooling strides to achieve spatial resolutions of 2x2 (Event-Slot-VLM (2x2)), 4x4, and 8x8.

Table 7 shows the results. Increasing the sampled frames or spatial resolutions improves performance but increases the number of video tokens and the computational cost. By default, we use 8 sample frames and a 4x4 spatial resolution to balance complexity (192 tokens in total) and performance.

Table 7: Ablation studies on sampling different number of temporal frames for the Object-Slots branch, and on using different spatial resolutions for the Event-Slots branch

| Model | | # video tokens | In-domain | | MSVD-QA | |
|---|---|---|---|---|---|---|
| | | | Acc. | Score | Acc. | Score |
| Object-Slots branch | Object-Slot-VLM (4 frames) | 32 | 42.1 | 2.55 | 72.4 | 3.65 |
| | Object-Slot-VLM (8 frames) | 64 | 46.5 | 2.69 | 73.1 | 3.71 |
| | Object-Slot-VLM (16 frames) | 128 | **46.8** | **2.7** | **74.3** | **3.72** |
| Event-Slots branch | Event-Slot-VLM (2x2) | 32 | 39.3 | 2.44 | 70.9 | 3.55 |
| | Event-Slot-VLM (4x4) | 128 | 47.1 | 2.67 | 73.1 | 3.67 |
| | Event-Slot-VLM (8x8) | 512 | **50.4** | **2.79** | **76.6** | **3.77** |

**Influence of Three Stages Training** Stage 1 aims to pre-train the slot attention modules (with the objective of reconstructing the input features) to facilitate the learning of object-centric and event-centric slot representations, i.e., the forming of semantically decomposed tokens/slots. Stage 2 separately trains the Object-Slots branch and the Event-Slots branch to facilitate the system focusing on the optimization of each branch separately, which eases the optimization. Stage 3 jointly optimizes the two branches and the projection layer. Table 8 shows the ablation study on the influence of each training stage. We have the following observations/conclusions. 1) If there is only one joint training stage as Model-1, the performance drops by 1.8% and 2.4% in accuracy on In-domain and MSVD-QA, respectively. When compared with other methods as shown in Table 1 on MSVD-QA, our Model-1 with one joint training stage achieves 72.5% in accuracy, ranks the second best among all other methods. It is still effective over many other methods but is less effective than our final scheme with three stages training. 2) Skipping either Stage 1 or Stage 2 leads to performance drop.

Table 8: Ablation study on the three-stage training on In-domain and MSVD-QA. "No" denotes that this stage training is not used while "Yes" denotes that this stage training is used. We report the accuracy (%) and score.

| Models | Stage 1 | Stage 2 | Stage 3 | In-domain | | MSVD-QA | |
|---|---|---|---|---|---|---|---|
| | | | | Acc. | Score | Acc. | Score |
| Model-1 | No | No | Yes | 46.9 | 2.69 | 72.5 | 3.67 |
| Model-2 | No | Yes | Yes | 46.1 | 2.68 | 74.6 | 3.76 |
| Model-3 | Yes | No | Yes | 44.2 | 2.64 | 73.5 | 3.69 |
| Model-4 (final) | Yes | Yes | Yes | **48.8** | **2.75** | **74.9** | **3.76** |

Table 9: Performance (accuracy) comparison on the subset of Egoschema.

| Method | ViperGPT[38] | Sevila[46] | Video-LLaVA[23] | mPLUG-Owl[45] | LLoVi[47] | Slot-VLM |
|---|---|---|---|---|---|---|
| Accuracy | 15.8 | 25.7 | 36.8 | 33.8 | 50.8 | **55.8** |

# F    Comparison with the State-of-the-Arts on Multi-choice datasets

We also evaluate our models on multi-choice QA benchmarks, including Egoschema[29], NExT-QA[43] and STAR[42].

**Egoschema.** For EgoSchema, we have tested the performance on the 500-question subset and Table 9 shows the results. Slot-VLM significantly outperforms Sevila by 30% and outperforms the comparable 7B model Video-LLaVA by 19% in accuracy.

**NExT-QA.** For NExT-QA, Table 10 shows our Slot-VLM is competitive to VideoChat2. We believe using more training videos like VideoChat2 (1.1 million) would further enhance performance.

**STAR.** For STAR, Table 11 shows the comparisons. Different from [14, 2, 46, 4], our model is tested in zero shot manner, without accessing STAR during training. Our Slot-VLM achieves competitive performance with generalizable models of Flamingo-9B and InternVideo. Note that InternVideo uses a much larger number of videos (12 million) than ours, whereas our model uses only 100K videos for training.

# G    More Visualization

## G.1    Visualization of Q-Former Attention Maps from BLIP2

BLIP2 [19] is a representative visual language pre-trained model that aggregates image features (from CLIP encoder) by a Q-Former into 32 learnable query tokens. Q-Former employs a 12-head cross-attention mechanism to aggregate visual features into a set of queries, which subsequently processes the queries by a self-attention module and a feed-forward layer. We analyze whether such well-trained Q-Former has learned decoupled semantics for the 32 query tokens. Figure 6 visualizes the learned cross-attention masks (laid out on the original image) corresponding to the 32 queries (32 columns, $\mathbf{q}_{h,1}, \mathbf{q}_{h,2}, \ldots, \mathbf{q}_{h,32}$) from 12 heads (shown in 12 rows, $h = 1, 2, \ldots, 12$) for two images in (a) and (b) respectively. We can see that a query focuses on different regions for different heads. For a head, usually the information was allocated into only a few queries. However, there is no obvious evidence that different queries have learned decoupled semantics. In contrast, as shown in Figure 3, our learned spatial slots have more remarkable decoupled semantics.

## G.2    Visualization of Q-Former Spatial Attention Maps from Spatial–QFormer-VLM

We visualize the spatial attention maps of the learned queries of Q-Former from our Spatial-QFormer-VLM. Q-Former uses cross-attention of 12 heads to aggregate the visual features. In Figure 7, we show the masks for 12 heads separately as shown in the $3 \times 4$ grids separated by green lines. For each head, we show all the 8 frames in different rows; the first column shows the original frame; the second to the ninth columns show the cross attention mask (from Q-Former) for the 8 queries. We can see that the queries are not clearly decoupled, with each one being a mixture of spatial features. A feature in a spatial position is usually allocated to a couple of queries instead of one. Compared with the spatial attention masks from our Slot-VLM as shown in Figure 3, it is less semantically decoupled for the learned queries.

Table 10: Performance (accuracy) comparison on NExT-QA.

| Dataset specific trained (In domain) | Accuracy |
|---|---|
| MIST[14] | 57.2 |
| GF(uns)[2] | 58.8 |
| Sevila[46] | 73.8 |
| Zero-shot (Generalization) | Accuracy |
| InternVideo[41] | 49.1 |
| Mistral(7B)[16] | 51.1 |
| VFC[30] | 51.5 |
| LLoVi[47] | 54.3 |
| MVU(13B)[33] | 55.2 |
| ViperGPT(GPT-3.5)[38] | 60.0 |
| LangRepo(12B)[18] | 60.9 |
| VideoChat2[21] | 61.7 |
| Slot-VLM | **62.0** |

Table 11: Performance (accuracy) comparison on STAR.

| Dataset specific trained (In domain) | Accuracy |
|---|---|
| MIST[14] | 51.1 |
| GF(uns)[2] | 53.9 |
| Sevila[46] | 64.9 |
| LRR[4] | 70.5 |
| Zero-shot (Generalization) | Accuracy |
| Flamingo-9B[1] | 41.8 |
| InternVideo[41] | 41.6 |
| Slot-VLM | **42.7** |

### G.3  Visualization of Spatial Attention Maps for Spatial Slots from Stage 1 and Stage 2

In the first stage pre-training, slot attention is learned by reconstructing the features. In the second stage, LLM is incorporated for video instruction tuning. Here, we take spatial slots as examples to compare the slot attention masks from stage 1 and stage 2. Figure 8 and Figure 9 show the visualization for two examples, respectively. Interestingly, we observe that after the instruction tuning, the learned slots are better decoupled, where a spatial position usually contributes to multiple slots in stage 1 but only contributes to a very few slots in stage 2. Similar phenomenons are obtained for temporal slots.

### G.4  Visualization of Learned Event-Centric Slots

Figure 10 shows the enlarged visualization of attention masks for the 13-th spatial position that is previously shown in Figure 4.

## H  Discussion on the Application on Long Video Understanding

For several-minute-long videos, Slot-VLM can still capture important objects and events information for video understanding. Experiments on the ActivityNet-QA dataset, where the average length of video clips is 111.6 seconds, and the maximum length is 285 seconds, show the superiority of our Slot-VLM over many other methods (see Table 1, note that the amount of instruction tuning data is much smaller than that in VideoChat2). For scenes with more objects, some similar objects would be gathered into the same slot. Even if not being perfect, our semantics decoupled representations presents its superiority to pooling-based and Q-former based strategies where semantics are mixed in the generated tokens for video understanding.

For much longer videos (e.g., hour-long videos), it would be difficult to represent the dynamic scenes with only a small set of event slots. We could extend our framework to well serve long video understanding by partitioning the long-video to chunks of shorter duration, where the object-centric and event-centric slots are generated for each chunk and these slots from chunks are concatenated sequentially as the input to LLM. We leave this as future work.

## I  Limitations

In this work, we introduce a new framework, Slot-VLM, that aims to generate a small set of event-centric video tokens to comply with LLM for effective video reasoning. Currently, we leverage slot attention to learn object-centric slots and event-centric slots. However, the learned slot representations are still not perfect where object instances are still not perfectly segmented and events are not perfectly partitioned. With such imperfect slot representations, our framework achieves the state-of-the-art performance, demonstrating strong potential. We believe the advancement of unsupervised object-centric representation learning in future could further enhance the performance of our framework.

For much longer videos (e.g., hour-long videos), it would be difficult to represent the dynamic scenes with only a small set of event slots. Our framework is extensible to support those very long videos but we have not implemented and experimented on such cases. For example, for a three-hour-long video, we could partition the video into chunks with each chunk being $\tau$ seconds and summarize dense video features into sparse $N_s$ object-centric and $N_f$ event-centric tokens for *each* chunk and concatenate the tokens of chunks sequentially

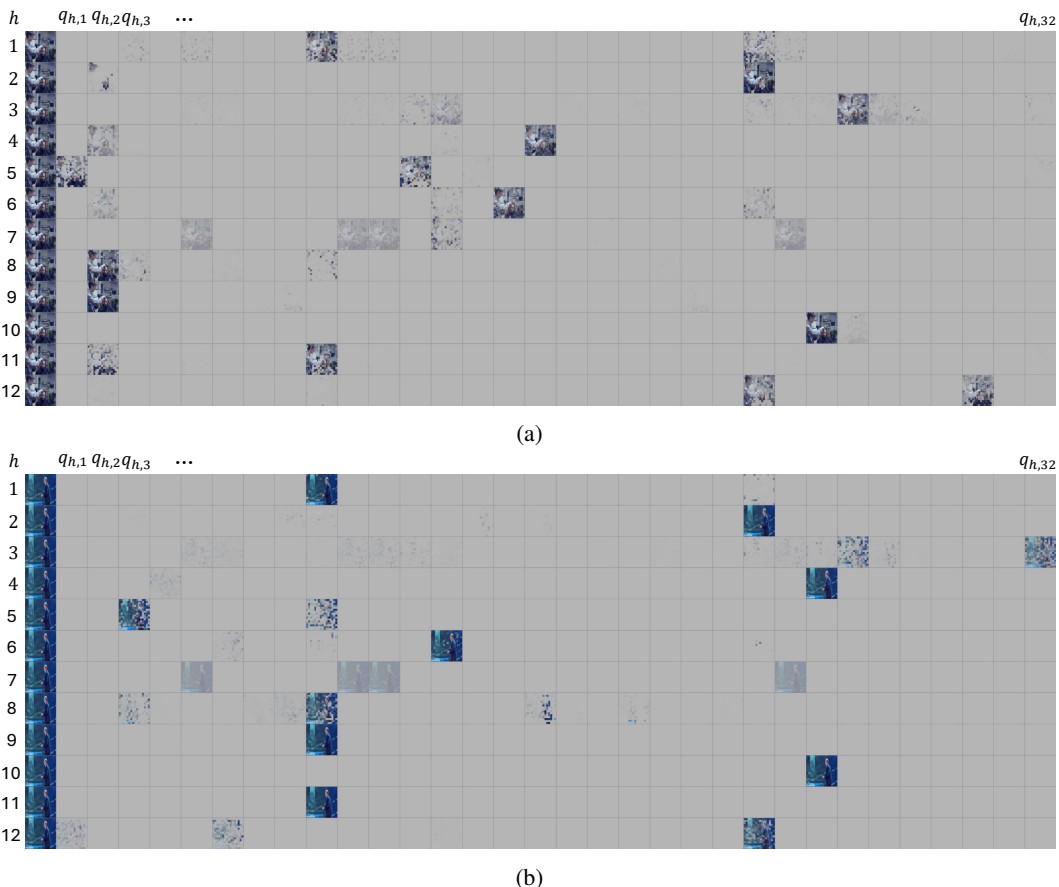

Figure 6: Visualization of spatial attention masks from the Q-Former in BLIP2 for two images in (a) and (b) respectively. We show the learned query masks for the 12 heads in 12 rows, respectively. In each row, we show the masks for the 32 queries. Note that the first column show the original image repeated by 12 times. There is no obvious evidence that different queries have learned decoupled semantics.

as the input to LLM for inference. This is feasible in practice by borrowing experience from LLaMA-VID [22], which is designed for long-form video understanding. LLaMA-VID encodes each frame into two tokens at a sample rate of 1 frame per second to support three-hour-long videos. In our framework, when we have $N_s + N_f = 192$ tokens for each chunk, by setting the duration of a chunk to $\tau = 96$ seconds, we equivalently have 2 tokens per frame on average, facilitating the handling of hour-long-video as LLaMA-VID does.

## J Impact Statements

This paper endeavors to make notable advancements in video-language modeling, a field integral to enhancing our interaction with and comprehension of video content. We aspire to drive positive innovation, yet we remain aware that technological progress may have unforeseen side effects. We call upon the users to use AI responsibly. When we release code and models, we would set a safeguard by requiring that users adhere to usage guidelines.

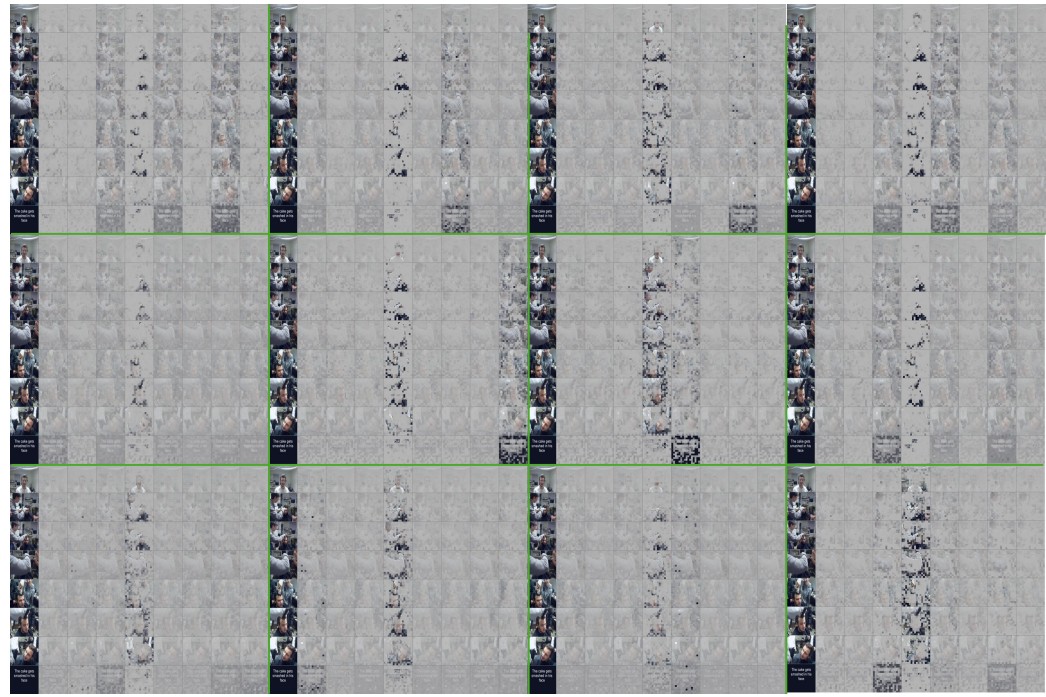

Figure 7: Visualization of spatial attention masks from our baseline scheme Spatial-QFormer-VLM for 8 frames of a video. Since there are 12 heads in Q-Former, we plot 12 ($3 \times 4$) sets of attention masks separated by green lines. For each head, we have $t = 8$ frames as shown in each row; the first column shows the original frame; the second to the ninth columns show the cross attention mask (from Q-Former) for the 8 queries. We can see that the queries are not clearly decoupled, with each one being a mixture of spatial features. A feature in a spatial position is usually allocated to a couple of queries. Compared with the spatial attention masks from our Slot-VLM as shown in Figure 3, it is less semantically decoupled for the learned queries.

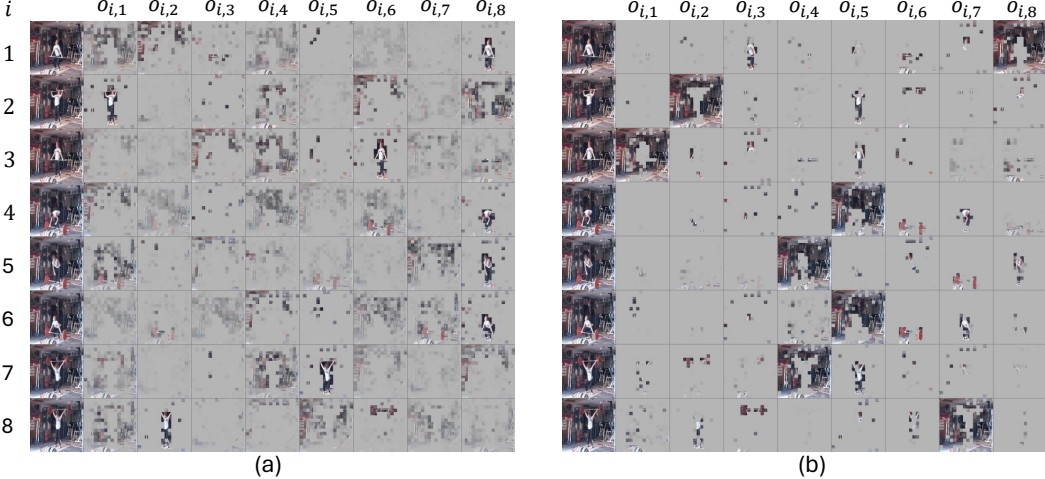

Figure 8: Visualization of spatial attention masks from (a) the stage 1 pre-training, and (b) the stage 2 after instruction tuning. We have $t = 8$ frames as shown in 8 rows, indexed by $i$, where $i = 1, \ldots, t$, respectively. The first column shows the original frame. The second to the ninth columns show the cross attention mask (from slot attention) for the $N_o = 8$ object-centric slots $\mathcal{O}_i = \{\mathbf{o}_{i,1}, \ldots, \mathbf{o}_{i,N_o}\}$. Interestingly, we can see that after the instruction tuning, the learned slots are much more decoupled, where a spatial position usually contributes to multiple slots in stage 1 but only contributes to a very few slots in stage 2.

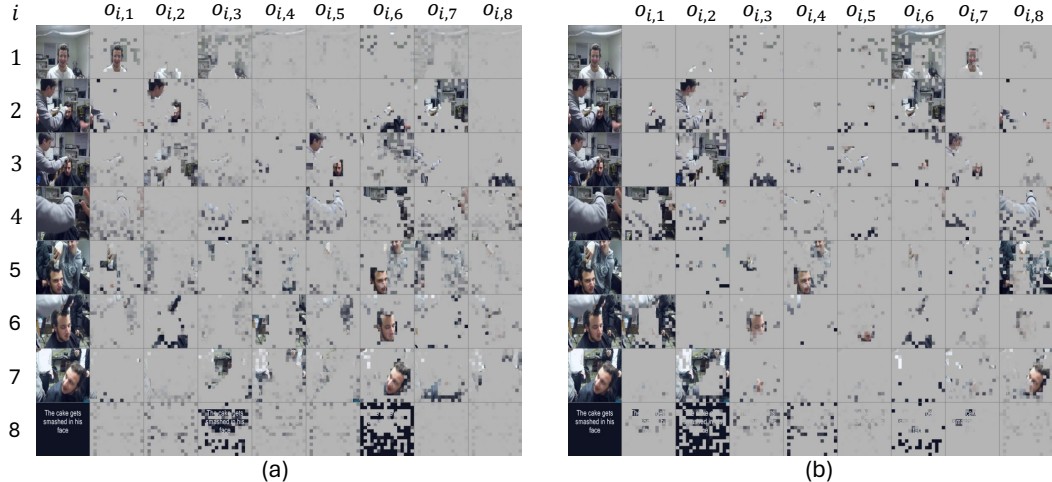

Figure 9: Visualization of spatial attention masks from (a) the stage 1 pre-training, and (b) the stage 2 after instruction tuning. We have $t = 8$ frames as shown in 8 rows, indexed by $i$, where $i = 1, \ldots, t$, respectively. The first column shows the original frame. The second to the ninth columns show the cross attention mask (from slot attention) for the $N_o = 8$ object-centric slots $\mathcal{O}_i = \{\mathbf{o}_{i,1}, \ldots, \mathbf{o}_{i,N_o}\}$. Interestingly, we can see that after the instruction tuning, the learned slots are much more decoupled, where a spatial position usually contributes to multiple slots in stage 1 but only contributes to a very few slots in stage 2.

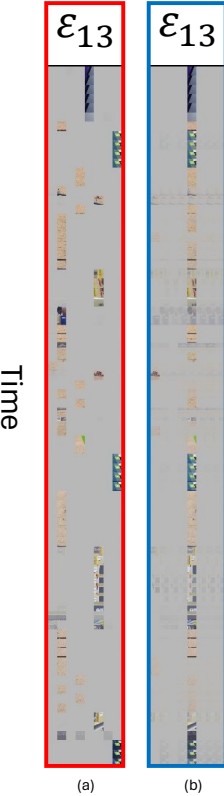

Figure 10: Enlarged visualization of temporal attention mask for the 13-th spatial position from (a) our Event-Slots branch and (b) Temporal-QFormer-VLM, respectively. For simplicity, we also refer to slot as query here. For the 13-th spatial position, we denote the set of learned temporal queries by $\mathcal{E}_{13}$. For this spatial position, the models generate $N_e = 8$ queries by aggregating the temporal visual tokens. The attention masks for $\mathcal{E}_{13}$ are denoted by a map of $T$ rows and $N_e = 8$ columns, with the visibility indicating which queries this temporal position belongs to. The higher the visibility, the greater the affinity between this temporal position and the query. We can see that in our Slot-VLM, similar contents tend to be allocated to the same slot, *i.e.*, different slots capture different contents (events) and present decoupled semantics. In contrast, in Temporal-QFormer-VLM, different contents are usually assigned to the same query or are uniformly assigned to different queries.

