# OpenReview forum: "Slot-VLM: Object-Event Slots for Video-Language Modeling"
_NeurIPS.cc/2024/Conference — NeurIPS 2024 poster_

### Official Review · Reviewer_KJVB · 2024-06-29

**Soundness:** 3
**Presentation:** 3
**Contribution:** 3
**Rating:** 6
**Confidence:** 4

**Summary:**

This paper proposes to use object-centric slot representations as visual representations in vision language models. Specifically, it adopts the popular video LLM paradigm and concatenates a pre-trained visual encoder with a pre-trained LLM. It uses a slot attention method to group visual features into object slots and event slots before feeding them into the LLM. These slot representations compress visual features into a small set of tokens and stand out in terms of efficiency. The final model, Slot-VLM, performs well in video question answering tasks and has the potential to handle long videos.

**Strengths:**

1. It's an intuitive and good idea to apply object-centric inductive bias to vision-language models.
2. The experiments in the main paper and appendix are sufficient, assessing various aspects of the method.
3. The performance is surprisingly good.

**Weaknesses:**

1. Are the temporal slots really event slots? Since there are m spatial positions after downsampling, what if an object moves around from one spatial patch to another? Then the event cannot be effectively captured by the slots, since the temporal slot attention happens in the fixed spatial position along the temporal dimension.
2. The visualization seems poor. It was expected that slot attention, as a learnable method, should performs better than naive clustering algorithms, as used in ChatUniV. However, the results shown in Figure 3 do not really group objects as expected. It still looks similar to the visualization of ChatUniV.
3. Regarding object-centric/slot-attention in videos, there are some related works that are not discussed. For example, <Object-Centric Learning for Real-World Videos by Predicting Temporal Feature Similarities, NeurlPS 2023>, <Unsupervised Open-Vocabulary Object Localization in Videos, ICCV 2023>.

**Questions:**

1. A related work, ChatUniV in CVPR 2024, uses clustering algorithm to merge visual tokens, thereby reduce computational cost. Slot attention can be regarded as an upgraded (learnable) clustering algorithm. Apart from this, what is the key difference between Slot-VLM and ChatUniV?
2. SlowFast network is initially proposed to address the video redundancy problem. While in Slot-VLM, the slot-attention seems to also reduce redundancy by aggregating the visual tokens into compact slot representations. In that case, is it still necessary to use SlowFast?

**Limitations:**

Please refer to weaknesses and questions above.
In summary, the object-centric inductive bias used in this work is interesting. It also shows potential on handling long-videos given the token compression strategy and the impressive performance on standard benchmarks. However, the claim of Object-Event Slots in the title is questionable. 1) The object-slots seem to not effectively group objects according to visualization. 2) Event-slots. See weakness point 1.

---

> ### Author Rebuttal · Authors · 2024-08-02
>
> Thank you very much for your positive feedback regarding the idea, sufficiency of experiments, and the surprisingly good performance. We greatly appreciate your insights and suggestions and will thoroughly incorporate them into our revised manuscript. Below, we provide detailed responses.
>
> **Q1**: Are the temporal slots really event slots? Since there are m spatial positions after downsampling, what if an object moves around from one spatial patch to another? Then the event cannot be effectively captured by the slots, since the temporal slot attention happens in the fixed spatial position along the temporal dimension.
>
> **A1**: Thank you for the insightful question. We understand the concern. Our intention with the term “event slots” is to conceptually align these slots with events at a high-level, as mentioned in lines 203-204 of the manuscript: “Temporal slots aggregate the temporal tokens for explaining parts of the input, similar to identifying events. Thus, we name the learned slots as event-centric slots.”
>
> We have considered this question during our design. To alleviate it, instead of learning event slots for each spatial position, we divide the feature map into 4x4=16 non-overlapped large spatial regions (downsampling), with each region corresponding to a large local region of 56x56 pixels when the frame resolution is 224x224. **This allows us to observe the temporal dynamics of a large local region (56x56 pixels), which helps infer the evolution of part of an event within that region.**
>
> We acknowledge that this method might not perfectly capture the entire object dynamic, but it allows for a partial understanding of object evolution within a large local region. Through leveraging the learned slots from all 16 large local regions, the LLM can achieve a more comprehensive perception of global temporal dynamics.
>
> The temporal slots aggregate semantically consistent temporal segments, akin to how spatial slots aggregate spatially consistent regions (object-centric slots). While the segments within a slot might not encompass a complete event, they are likely part of the same event. We will revise the manuscript to clarify this design consideration and improve the explanation of how event slots work in our model.
>
>
> **Q2**: The visualization seems poor. It still looks similar to the visualization of ChatUniV. What is the key difference between Slot-VLM and ChatUniV?
>
> **A2**: Thank you for your insightful question. Yes, the visualization of slot attention is still not very satisfactory. Actually, the unsupervised object-centric representation learning remains a challenging task in the field.
>
> Our Slot-VLM significantly outperforms ChatUniV (see Table 1) due to two main reasons.
>
> 1) **Slots can preserve more original information than clustering centers.** Slot attention is trained to reconstruct the original features from slots. In contrast, ChatUniV uses the average token within each cluster to represent the corresponding cluster, where the averaging can result in a loss of fine-grained details, whereas the detailed visual information is crucial for accurate video understanding and reasoning.
>
> 2) **Slots are learnable and can be jointly optimized within the entire framework, whereas clustering (being not learnable) in ChatUniV is task-agnostic.** Our ablation study below shows that freezing the slot attention subnetworks at Stage 3 in our framework leads to a noticeable drop in performance when compared with our final scheme with slot attention jointly tuned (see Table F).  Note that we will add the results that freezing the slot attention subnetworks at Stage 2 and Stage 3 once we obtain, which we guess they are lower than that freezing slot attention subnetworks only at Stage 3.
>
> Table F: Ablation study on the influence of end-to-end optimization of the slot attention modules. Slot-VLM (slot frozen) denotes the slot attention modules are frozen in Stage 3.
>
> | Models                 | In-domain (Acc./Score) | MSVD-QA (Acc./Score) |
> |:----------------------:|:----------------------:|:--------------------:|
> | Slot-VLM (slot frozen) |     46.2/2.68                   |   74.0/3.74                   |
> | Slot-VLM               | 48.8/2.75              | 74.9/3.76            |
>
>
> **Q3**: Add related works regarding object-centric/slot-attention in videos.
>
> **A3**: Thank you for the helpful suggestion and we will add the following discussion.
>
> Some works have also explored object-centric learning in videos. VideoSAUR [A] extends the method of DINOSOUR [27] to video-based object-centric learning by incorporating temporal feature similarity loss, which introduces a motion bias for each frame. Fan et al. [B] propose an unsupervised method for open-vocabulary object localization in videos, where they run only once slot-attention on the entire spatial-temporal features to obtain video slots for all frames.
>
> In contrast, we leverage slot attention to learn semantically decoupled tokens as the input to LLMs and investigate the effectiveness of aligning these ‘concept tokens’ with LLM input.
>
> (Note [A] and [B] denote the referred two papers, respectively.)
>
>
> **Q4**: Is it still necessary to use SlowFast?
>
> **A4**: Yes, we had demonstrated the necessity of our two-branch design in line 309-320 of our manuscript. Table G shows that directly extending slot learning to spatial and temporal features increases memory and computation requirements, and the optimization difficulty.
> Table G: Ablation study on the effectiveness of joint spatial-temporal slots learning vs. our two-branch design. The FLOPs and number of parameters for the reduced token generation module are presented.
>
> | Model          | In-domain (Acc.) | MSVD-QA (Acc.) | #FLOPs (G) | #Param. (M) |
> |:--------------:|:---------:|:-------:|:----------:|:-----------:|
> | Slot-Joint-VLM | 46.7      | 72.8    | 1304.8     | 13.7        |
> | Slot-VLM      | 48.8      | 74.9    | 41.6       | 27.3        |

---

> > ### Comment · Reviewer_KJVB · 2024-08-08
> > **Response of rebuttal and further questions**
> >
> > I appreciate the detailed rebuttal from the author.
> >
> > Regarding the definition of the event slots, please ensure to add the clarification into the paper, as this definition is somehow misleading.
> >
> > As for the poor visualization result, I understand that object-centric learning methods struggle to handle real-world scene. But some related works, e.g., <Bridging the Gap to Real-World Object-Centric Learning> already shows decent object grouping results on real-world data. Therefore, I'm still a bit confused about the visualization. If the visualization is not good, then the object-slot and event-slot definition is a bit more questionable.
> >
> > For the conceptual comparison with Chat-UniV, the authors mentioned that "Slots can preserve more original information than clustering centers." and "Chat-UniV ... averaging can result in a loss of fine-grained details". As far as I know, slot-attention module usually uses a very compact slot representation with low embedding dimension to form a information bottleneck. I'm curious about the implementation of the slot attention in the paper. What is the dimension? How do you use the slots to input to the LLM? Why does it preserve more abundant information than average pooling visual tokens?
> >
> > And the last point about SlowFast and Slot-Joint-VLM, since there would be a hyper-parameter to define the number of slots in your model, I wonder if you define the total number of slots as the same in the two settings in Table G, what brings the significant difference in the FLOPs?
> >
> > Thanks.

---

> > > ### Author Response · Authors · 2024-08-08
> > > **Response to the questions**
> > >
> > > Thank you very much for your insightful questions and constructive suggestions, which are very helpful in improving our work.  We will add the clarification related to event slots into the revised manuscript.
> > >
> > > **About the visualization**: As far as we know, DINOSAUR [27] as you referred, which learn the object-centric representation on pre-trained feature space (e.g., DINO, CLIP features) by reconstructing the features, was the state-of-the-art solution on real world data to obtain object grouping results. As the visualization shown in [27], the performance also depends on the complexity of the image contents. It can work well on simple datasets such as MOVi-C, MOVi-E (see Fig. 2 in [27]) but is still not perfect on more complicated images (e.g., COCO). We use the same technique of DINOSAUR to learn slots by reconstructing pre-trained CLIP features. Even though the visualization is still not good, we can observe the learned slots tend to group the features of similar semantics. Our Slot-VLM achieves superior performance, demonstrating strong potential. We believe that more advanced object-centric learning method would further enhance the performance of our framework, and we are eager to have a try if we can use a better object-centric learning method.
> > >
> > > [27] Bridging the Gap to Real-World Object-Centric Learning, ICML'23.
> > >
> > > **About preservation of information**: Thank you for the good question. The dimension of a slot is 1024. Slot-attention module generates compact slot representations, where the number of output slots is much smaller than the number of input tokens. Actually, the embedding dimension is **not** low.  Take the Object-Slots branch as an example. For a video frame with feature of 16x16x1024 (height x width x dimension), i.e., 16x16=256 tokens, slot attention taken the 256 tokens as input to generate 8 slots with each of 1024 dimensions. After going through the projection layers, the slots for t (t=8) frames are sequentially concatenated to have 64 slots, as the input to LLM, with each slot as a token.
> > >
> > > Slot attention module is optimized to enable slots to reconstruct the input as better as possible, even though the bottleneck/compression would lead to loss of some information. Each slot is a weighted summarization of the input. In contrast, the forming of clustering center (using pooling/averaging operation) is not optimizable to preserve information. Therefore, at the same number of output tokens, slots are able to preserve more information.
> > >
> > > **About SlowFast and Slot-Joint-VLM**: Yes, the total number of slots are the same. For the cross-attention operation in slot attention module, we know the complexity is proportional to the number of Key/Value (i.e., number of input tokens). Our decomposition of the spatial-temporal tokens to spatial and temporal enables the slot-attention to operation along each dimension (spatial or temporal), which greatly reduced the number of Key/Value (input tokens) to infer slots. In addition, we have conducted temporal down-sampling in the spatial branch (Object-Slots branch) while preserving the spatial resolution. Similarly, we have conducted spatial down-sampling in the temporal branch (Event-Slots branch) while preserving the temporal resolution. This further reduces the complexity. We will add more analysis to our revision.
> > >
> > > Thank you again for your great efforts to make our paper clear and solid. Any questions/comments are warmly welcomed!

---

### Official Review · Reviewer_wQQx · 2024-07-11

**Soundness:** 2
**Presentation:** 3
**Contribution:** 3
**Rating:** 4
**Confidence:** 4

**Summary:**

This paper aims to learn a higher level of abstract representation as the input tokens to LLM. The paper proposes a dual-branch (object- and event-centric) to extract the concepts and uses the three-stage training paradigm. The proposed Slot-VLM is evaluated on three VQA datasets and has shown state-of-the-art performance.

**Strengths:**

- The idea of decomposing the semantics of input tokens to LLM is interesting and crucial for human-like reasoning.
- The evaluation results show strong performance on three datasets (Table 1) and better efficiency (4.4), which validates that such structured representations can improve the performance of VQA.
- The extensive ablation study (Section D) is meaningful and comprehensive enough.
- The details of reproducing the method are comprehensive.

**Weaknesses:**

My main concern is that the authors claim the temporal branch of Slot-VLM as the event branch. The slot attention focuses on local regions temporally, which makes it only capture low-level local motion instead of actual events with high-level semantics. Moreover, the visualization of temporal attention masks (Figures 8 and 10) can barely provide cues that the model can learn events-centric representation (L53, L55, L65). Thus, I believe the event-centric slot is overclaimed in the entire article. I hope the authors can clarify this.

**Questions:**

See weakness. The claim of event-centric slot is my main concern.

**Limitations:**

Yes, the authors raise the limitations of this paper in two directions: imperfect concept discovery and extension for hours-long videos.

However, it would enhance the paper much more if the authors could provide more insights into what other semantic concepts the proposed model lacks, which would complement the existing object- and event-centric representations (Although I am not convinced the model actually captures the “event” semantic)

---

> ### Author Rebuttal · Authors · 2024-08-02
>
> Thank you very much for your positive feedback and helpful suggestions. We greatly appreciate your recognition of the interestingness and cruciality of the idea, strong performance, meaningful and comprehensive ablation study, and comprehensive reproducibility details. We have carefully considered your valuable suggestions and comments and will incorporate them into our revised manuscript. Please find our detailed responses below.
>
> **Q1**: My main concern is that the authors claim the temporal branch of Slot-VLM as the event branch. The slot attention focuses on local regions temporally, which makes it only capture low-level local motion instead of actual events with high-level semantics. Moreover, the visualization of temporal attention masks (Figures 8 and 10) can barely provide cues that the model can learn events-centric representation (L53, L55, L65). Thus, I believe the event-centric slot is overclaimed in the entire article. I hope the authors can clarify this.
>
> **A1**: Thank you for your constructive comments. We will revise the descriptions for clarity. Our intention is to conceptually align event slots with events at a high level, as defined in lines 203-204 of the manuscript: "**Temporal slots aggregate the temporal tokens for explaining parts of the input, similar to identifying events. Thus, we name the learned slots as event-centric slots.**" More precisely, we leverage temporal slots to aggregate semantically consistent temporal segments, akin to how spatial slots aggregate spatially consistent regions, termed object-centric slots. While these temporal segments within a slot might not form a complete event, they are likely part of the same event.
>
> **About local region**: We understand your concern regarding the focus on local regions. This was a careful consideration in our design process. To alleviate this, instead of learning event slots for each spatial position along the temporal dimension, we divide the feature map into 4x4=16 non-overlapped spatial regions (downsampling), where each region corresponds to a large local region of 56x56 pixels when the frame resolution is 224x224. Observing the temporal dynamics of such a large local region (56x56 pixels) allows for partial inference of event evolution within the region, although it may not be perfect. Note that the temporal aggregation through slots is performed on CLIP features, which possess high-level semantics, thus enhancing the slots’ ability to capture high-level semantics. Moreover, by utilizing the learned slots for all 16 large local regions, the LLM can achieve a global perception of temporal dynamics.
>
> **About visualization**: We recognize that the current technique using slot attention for forming objects (spatial) and action/event (temporal) segmentation are not yet satisfactory. In Figure 10 (a), the good news is that similar contents tend to be allocated to the same slot and different slots capture different contents. The patterns from slots (Figure 10 (a)) are much better than those obtained from Q-Former (see Figure 10 (b)). We acknowledge that our work is still far from forming idea events, but we have taken a small step towards that goal. The superior performance demonstrates the strong potential of our proposed idea of decomposing the semantics of video tokens to align with LLM. More joint efforts from the community are still needed to push this field forward.

---

> ### Comment · Reviewer_wQQx · 2024-08-13
>
> I thank the authors for their response.
> My main concern about the clarity of "event slots" has not been addressed well, which is the major claim of the paper. As mentioned by reviewer KJVB, the paper only considers "events" that represent the temporal change within a patch. This would limit the proposed method to a simple domain of activities. For example, such a definition of event slots would not be able to handle a scenario where a subject manipulates objects with hands (e.g., Assembly-101 and Ego4D) or kitchen scenarios (e.g., Epic-kitchen) where the subjects and their hands move consistently. I would suggest that the authors revise the claim of the event slot as the term can be misleading.
> Moreover, from the rebuttal by the authors "slots is performed on CLIP features, which possess high-level semantics, thus enhancing the slots’ ability to capture high-level semantics" is not very convincing. The statement should be substantiated with some qualitative results that can demonstrate the event slots actually capture high-level semantics, as the current ones do not really show meaningful semantics.

---

> > ### Author Response · Authors · 2024-08-14
> >
> > Thank you very much for your great efforts and valuable feedback. We will follow your advice to revise the claim of the event slot to more accurately reflect the capabilities and limitations. Particularly, we will reflect its functioning on "large local regions (e.g., patches of 56x56 pixels)" and capturing of portions of events.
> >
> > We appreciate your constructive feedback, which contributes a lot to the improvement of our work!
> >
> > Thank you!

---

### Official Review · Reviewer_gzPw · 2024-07-12

**Soundness:** 3
**Presentation:** 3
**Contribution:** 3
**Rating:** 6
**Confidence:** 4

**Summary:**

This paper introduces a new framework called Slot-VLM that aims to aggregate spatial and temporal relationships between frames for more effective video understanding with Large Language Models (LLM). The Slot-VLM approach aims to generate video tokens that are semantically disentangled for better alignment with the frozen LLM. Specifically, the paper proposes the Object-Slots module to extract object-centric slots from high spatial resolution features that are sampled with a low frame rate. Additionally, it also introduces the Event-Slots module, which helps to aggregate the temporal dynamics between frames by learning event-centric slots from features with low spatial resolution but sampled at a high frame rate. The authors demonstrate the benefits of their proposedSlot-VLM approach by evaluating across three open-ended video question-answering datasets including MSVD-QA and ActivityNet-QA, where it outperforms existing video-language models by a significant margin.

**Strengths:**

1) The model figures are informative and especially helpful in helping the reader to understand the different stages of the learning algorithm as well as the intuition behind each stage. The slot attention approach, from the aggregation of video tokens to the combination of object and events-centric slots as well as their input to the LLM, is described well.

2) While the introduced Slot-VLM bears strong similarities to existing work on hierarchical understanding of videos such as the SlowFast approach, the concept of generating semantically-disentangled video tokens to align with frozen LLMs is relatively interesting and novel. In contrast to existing video-language models which often rely on pooling or compressing information into learnable queries, the ability to generate interpretable attention maps is particularly helpful to visualize what the model is focusing on.

3) In this paper, the authors conduct comprehensive comparisons with state-of-the-art video-language models. These evaluations highlight the limitations of existing work under relatively fair settings of using the same visual and language models as well as similar training data. The results also further emphasize the improvements brought by Slot-VLM.

**Weaknesses:**

1) The proposed Slot-VLM approach appears to be much more effective at aggregating disentangled spatial and temporal relationships between frames, as evidenced by the performance gap between its achieved results and those of Video-ChatGPT and Video-LLaVA in Table 1. However, it is unclear how much of the performance gain is also due to the final number of video tokens that are passed into the underlying LLM for reasoning and generation. It would be helpful to include a comparison of the different number of video tokens used among the different video-language models as well as the number of trainable parameters during each of the training stages.

2) It is also unclear how scalable such an approach will be in handling much longer videos. Currently, it appears that 8 frames are used for the object-slots module. Along with using 8 event-centric slots in total across shorter videos, this may limit the performance of the proposed Slot-VLM approach on much longer videos, such as those used in the EgoSchema evaluation benchmark. Similarly, is there also a more efficient way to aggregate spatiotemporal relationships over more frames beyond just increasing the number of object and event-centric slots since this will increase the computational demand in the LLM?

3) It may be beneficial to the reader to include additional ablation experiments and analysis over the low frame sampling rate used in the Object-Slots Branch as well as the stride used for pooling across the spatial and temporal axes in the Event-Slots Branch. Currently, these hyperparameters appear to be selected based on final downstream performance but there is hardly any discussion on them.

**Questions:**

Please see above-mentioned limitations.

**Limitations:**

Yes.

---

> ### Author Rebuttal · Authors · 2024-08-02
>
> Thank you very much for your constructive suggestions, and positive feedback on the novelty and interestingness of our proposed concept, the ability to generate interpretable attention maps, the comprehensive comparisons, and the clarity of presentation. The detailed responses are provided below.
>
> **Q1**: Include a comparison of the different number of video tokens used among the different video-language models as well as the number of trainable parameters during each of the training stages.
>
> **A1**: We will include these in our revision.  Table B below shows the performance gain is not due to the final number of video tokens, where our Slot-VLM uses only 192 tokens that is much smaller than some other models.
>
> Table B: The number of video tokens used. ‘Varied length’ denotes the number varied with the video length. ‘-’ denotes we did not find information from their papers and will investigate their code in future to fill.  * denotes the likely number based on the paper’s description.
>
> | Model                    | Number of video tokens |
> |:------------------------:|:-----------------:|
> | Video LLaMA              | 256               |
> | Video Chat               | -                 |
> | Video-ChatGPT            | 356               |
> | Chat-UniVi               | Varied length     |
> | Video-LLaVA              | -                 |
> | Video-LLaVA$^†$  | -                 |
> | BT-Adapter               | 256               |
> | LLaMA-VID                | Varied length     |
> | VideoChat2               | 1536*         |
> | MovieChat                | -                 |
> | Slot-VLM (Ours)          | 192               |
> | Slot-VLM$^†$ (Ours) | 192           |
>
>
> Different models have different numbers of training stages and trainable parameters per stage. Most papers lack detailed information. For simplicity, we present the trainable parameters for each stage of Slot-VLM and the likely numbers for VideoChat2 based on the paper's description in Table C. Our Slot-VLM uses far less trainable parameters than VideoChat2 and requires less computing resource in training.
>
> Table C: The numbers trainable parameters (M: million; B: billion) in each stage for VideoChat2 and our Slot-VLM.
>
> |    Stage    | Stage 1 | Stage 2 | Stage 3 |
> |:-----------:|:-------:|:-------:|:-------:|
> | VideoChat2  |   101M  |   492M  |    7B   |
> | Slot-VLM    |   27M   |   37M   |   33M   |
>
> **Q2**: It is also unclear how scalable such an approach will be in handling much longer videos.
>
> **A2**: **Our approach is directly applicable to videos of a few minutes**. We have validated our framework on ActivityNet-QA (see Table 1), where the average length of video clips is 111.6 seconds, and the maximum length is 285 seconds, which is comparable to the reviewer referred EgoSchema (180 seconds).
>
> We also tested a 500-question subset of EgoSchema. Table D shows that our Slot-VLM achieves superior performance, which outperforms Video-LLaVA by 7% in accuracy.
>
> Table D: Performance (accuracy) comparison on the subset of Egoschema.
> | Method | ViperGPT | Sevila | Video-LLaVA | mPLUG-Owl | LLoVi | Slot-VLM |
> |:------:|:------:|:------:|:------:|:-----:|:------:|:------:|
> | Accuracy | 15.8   | 25.7   | 36.8     | 33.8  | 50.8 | **55.8**     |
>
> **For handing even longer videos, our framework can be extended.**  For example, for a three-hour video, we could partition the video into chunks, each of $\tau$ seconds, summarizing dense video features into $N_s$ object-centric and $N_f$ event-centric tokens for **each** chunk. These tokens can then be concatenated sequentially as input to LLM for inference.  At the setting of $N_s+N_f=192$ tokens per chunk and $\tau=96$ seconds (1fps) duration per chunk, we equivalently have 2 tokens per frame on average, facilitating hour-long video handling. Note that LLaMA-VID (Li et al., 2023b) also encodes each frame into two tokens but overlooks the exploration of temporal correlation (where ours outperforms LLaMA-VID in three benchmarks in Table 1). This strategy ensures that the total number of video tokens is proportional to the video length in a manageable and tolerable way. More efficient strategies for aggregating spatiotemporal relationships over longer videos will require further research and future efforts.
>
>
> **Q3**: Additional ablation over the low frame sampling rate used in the Object-Slots Branch as well as the stride used for pooling in the Event-Slots Branch.
>
>  **A3**: We conducted ablation studies using single branch settings. For the Object-Slots branch, we tested three frame sampling rates: 4 frames (Object-Slot-VLM (4 frames)), 8 frames, and 16 frames per video. For the Event-Slots branch, we tested four pooling strides to achieve spatial resolutions of 2x2 (Event-Slot-VLM (2x2)), 4x4, and 8x8.
>
> Table E shows the results. Increasing the sampled frames or spatial resolutions improves performance but increases the number of video tokens. By default, we use 8 sample frames and a 4x4 spatial resolution to balance complexity (192 tokens in total) and performance.
>
> Table E: Ablation studies on sampling different number of temporal frames for the Object-Slots branch, and on using different spatial resolutions for the Event-Slots branch.
>
> | Model                       | # Video tokens | In-domain (Acc./Score) | MSVD-QA (Acc./Score) |
> |:---------------------------:|:--------------:|:----------------------:|:--------------------:|
> | Object-Slot-VLM (4 frames)  | 32             |     42.13/2.55            |   72.36/3.65      |
> | Object-Slot-VLM (8 frames)  | 64             | 46.5/2.69              | 73.1/3.71            |
> | Object-Slot-VLM (16 frames) | 128            |  46.82/2.70          |      74.29/3.72     |
> | Event-Slot-VLM (2x2)        | 32             |       39.28/2.44          |    70.86/3.55       |
> | Event-Slot-VLM (4x4)        | 128            | 47.1/2.67              | 73.1/3.67            |
> | Event-Slot-VLM (8x8)        | 512            |   50.35/2.79          |     76.56/3.77     |

---

> ### Author Response · Authors · 2024-08-07
> **Citations in response A2**
>
> For the schemes in our response A2, the paper for ViperGPT is "ViperGPT: Visual Inference via Python Execution for Reasoning".
>
> The paper for LLoVi is "A Simple LLM Framework for Long-Range Video Question-Answering".
>
> The paper for mPLUG-Owl is "mPLUG-Owl: Modularization Empowers Large Language Models with Multimodality".

---

> > ### Comment · Reviewer_gzPw · 2024-08-13
> >
> > Thank you very much for your comprehensive efforts in addressing my concerns. In particular, I find your response on extending the proposed approach to much longer videos insightful. I also appreciate your efforts on the additional ablation experiments, given the time and computational constraints on your end, as well as providing more evaluation results on additional QA benchmarks. Thus, I will retain my original rating.

---

> > > ### Author Response · Authors · 2024-08-14
> > > **Thank you**
> > >
> > > We are very grateful for your valuable suggestions and feedback! Incorporating these results and clarifications are very helpful to make this paper more solid and comprehensive.  Thank you very much for your great efforts and time!

---

### Official Review · Reviewer_jNDE · 2024-07-13

**Soundness:** 3
**Presentation:** 3
**Contribution:** 3
**Rating:** 6
**Confidence:** 5

**Summary:**

This paper proposed Solt-VLM, which aims to generate a small set of semantically decoupled video tokens to comply with LLM for effective video reasoning. It design a dual-branch Object-Event Slots module to learn object-centric slots and event-centric slots to jointly capture the spatial object details and temporal dynamics. Experimental results demonstrate the superiority of our framework and show the effectiveness of using semantics-decoupled representations for aligning with LLM.

**Strengths:**

Overall, the main contribution of this paper is introducing slot attention as the multimodal connector between LLM and visual encoders. Even though it is more like a solid technical implementation based on previous works, I think it is a good try on the video-language domain and provides more insights into spatial-temporal modeling in MLLM. Also, the paper writing is good and the main motivation makes sense, the results show both efficiency and effectiveness.

**Weaknesses:**

- The proposed Slot-VLM is tested on open-ended MSRVTT-QA, MSVD, and ActivityNetQA with automatic ChatGPT-based metrics. Although the metrics is applied by some previous works, I don’t think this is a stable, reliable, and explainable enough evaluation which may vary a lot across GPT versions (check conclusion in [1]) and may generate different responses even for the same input.
- Also MSRVTT-QA/MSVD/ActivityNetQA is not designed from an object interaction perspective, which may not be effective enough for validating the object-centric design.
- Based on the previous two points, I would suggest testing on recent multi-choice QA benchmarks, including STAR [2] / NExT-QA [3] / Egoschema [4], which are annotated with clear answer choices, and designed from a human-object interaction view. And compared with recent model works like Sevila [5],  MIST [6], LRR [7], GF [8], which are also based on query-former, or iteratively attention on regions in the video.

[1] FreeVA: Offline MLLM as Training-Free Video Assistant, Arxiv24.
[2] Star: A benchmark for situated reasoning in real-world videos, NeurIPS 2021.
[3] Next-qa: Next phase of question-answering to explaining temporal actions, CVPR 2021.
[4] Egoschema: A diagnostic benchmark for very long-form video language understanding,  NeurIPS 2023.
[5] Self-chained image-language model for video localization and question answering, NeurIPS 2023.
[6] Mist: Multi-modal iterative spatial-temporal transformer for long-form video question answering, CVPR 2023.
[7] Look, Remember and Reason: Grounded reasoning in videos with language models, ICLR 2024.
[8] Glance and focus: Memory prompting for multi-event video question answering, NeurIPS 2023.

**Questions:**

please see weakness

**Limitations:**

authors provide limitation discussion in the paper

---

> ### Author Rebuttal · Authors · 2024-08-02
>
> We sincerely appreciate your positive feedback on our insights into spatial-temporal modeling in MLLM, the main motivation, and the effectiveness of method, as well as the quality of our writing. We have carefully considered your valuable comments and suggestions and are committed to incorporating them into our revised manuscript. Please find our detailed responses below.
>
> **Q1**: The proposed Slot-VLM is tested on open-ended MSRVTT-QA, MSVD, and ActivityNetQA with automatic ChatGPT-based metrics. Although the metric is applied by some previous works, I don’t think this is a stable, reliable, and explainable enough evaluation which may vary a lot across GPT versions (check conclusion in [1]) and may generate different responses even for the same input.
>
> **A1**: We acknowledge your comment and will include the comparisons on multi-choice QA benchmarks in our manuscript (see Response A3 below).
>
>
> **Q2**: Also MSRVTT-QA/MSVD/ActivityNetQA is not designed from an object interaction perspective, which may not be effective enough for validating the object-centric design.
>
> **A2**: Thank you for your comment and suggestion. We explored additional datasets for validation as discussed in Response A3 below.
>
> Note that our framework is expected to generally work well, since in general each image in a video is a capture of the world consisting of objects and backgrounds, where object-centric representations can be considered as the basic units. This can also be intuitively explained by human visual reasoning where object representation acts as a basic and fundamental representation. For humans, visual signals are processed initially through the primary visual cortex (V1) and subsequently integrated into higher-level visual processing areas, resulting in the formulation of complex visual object representations. Such high-level object representations together with brain-stored knowledge are then used for logical reasoning in brain. Similarly, our Slot-VLM generates object-centric and event-centric representations to provide the high-level vision source for effective LLM reasoning, where object-centric design would be generally helpful. The strong performance of our Slot-VLM (Table 1) also demonstrated the effectiveness of our design.
>
> In addition, for the three datasets, we found that there are plenty of video-question pairs that are related to querying the states/actions of objects and relations. For example, what is a dog doing? What’s the shape of the table? What are the animals that appear in the video? What is behind the person sitting in the video? Our object-centric design provides an effective bridge for visual and language modeling.
>
>
> **Q3**:  Based on the previous two points, I would suggest testing on recent multi-choice QA benchmarks, including STAR [2] / NExT-QA [3] / Egoschema [4], which are annotated with clear answer choices, and designed from a human-object interaction view. And compared with recent model works like Sevila [5], MIST [6], LRR [7], GF [8], which are also based on query-former, or iteratively attention on regions in the video.
>
> **A3**: Thank you for the helpful suggestions and we will add the comparisons in our revision.
>
> For EgoSchema, we have tested the performance on the 500-question subset and Table A-1 shows the results. Slot-VLM significantly outperforms Sevila by 30% and outperforms the comparable 7B model Video-LLaVA by 19% in accuracy.
>
> Table A-1 Performance (accuracy) comparison on the subset of Egoschema.
> | Method   | ViperGPT[9] | Sevila | Video-LLaVA | mPLUG-Owl | LLoVi [10] | Slot-VLM |
> |:--------:|:------:|:------:|:-----------:|:---------:|:--------:|:--------:|
> | Accuracy | 15.8   | 25.7   | 36.8        | 33.8      | 50.8 | **55.8**     |
>
> [9] ViperGPT: Visual Inference via Python Execution for Reasoning
>
> [10] A Simple LLM Framework for Long-Range Video Question-Answering
>
>
> For STAR, Table A-2 shows the comparisons. The referred four works [5-8] report the in-domain accuracy where training is performed on STAR. In contrast, without accessing STAR during training, our model is tested in zero shot manner. Our Slot-VLM achieves competitive performance with generalizable models of Flamingo-9B and InternVideo. Note that InternVideo uses a much larger number of videos (12 million) than ours, whereas our model uses only 100K videos for training.
>
> Table A-2 Performance (accuracy) comparison on STAR.
> |                            |      |
> |:--------------------------:|:----:|
> | **Dataset specific trained (In domain)**   |      |
> | MIST [6]                       | 51.1 |
> | GF(uns) [8]                   | 53.9 |
> | Sevila [5]                    | 64.9 |
> | LRR [7]                       | 70.5 |
> | **Zero-shot (Generalization)** |      |
> | Flamingo-9B                | 41.8 |
> | InternVideo                | 41.6 |
> | Slot-VLM                   | **42.7** |
>
> For NExT-QA, Table A-3 shows our Slot-VLM is competitive to VideoChat2. We believe using more training videos like VideoChat2 (1.1 million) would further enhance performance.
>
> Table A-3 Performance (accuracy) comparison on NExT-QA.
> |                            |      |
> |:--------------------------:|:----:|
> | **Dataset specific trained (In domain)**   |      |
> | MIST [6]| 57.2 |
> | GF(uns) [8]| 58.8 |
> | Sevila [5]| 73.8 |
> | **Zero-shot (Generalization)** |      |
> | InternVideo [11]| 49.1 |
> | Mistral(7B) [12]| 51.1 |
> | VFC [13]| 51.5 |
> | LLoVi [10]| 54.3 |
> | MVU(13B) [14]| 55.2 |
> | ViperGPT(GPT-3.5) [9]| 60.0 |
> | LangRepo(12B) [15]| 60.9 |
> | VideoChat2 [16]| 61.7 |
> | Slot-VLM| **62.0** |
>
> [11] InternVideo: General Video Foundation Models via Generative and Discriminative Learning
>
> [12] Mistral 7B
>
> [13] Verbs in Action: Improving verb understanding in video-language models
>
> [14] Understanding Long Videos in One Multimodal Language Model Pass
>
> [15] Language Repository for Long Video Understanding
>
> [16] MVBench: A Comprehensive Multi-modal Video Understanding Benchmark

---

> > ### Comment · Reviewer_jNDE · 2024-08-08
> > **Thanks for your detailed response and experiments**
> >
> > Thanks authors for their every effort to provide more results on multi-choice QA benchmarks/datasets.
> >
> > Happy to see the proposed Slot-VLM achieves comparable performance with limited resources.
> > Overall, I would highly suggest including that multi-choice exp as a main result in the final version, considering it is a more stable/interpretable evaluation and can provide more grounded hints for further work.
> >
> > On the other hand, I would also suggest including a more inclusive comparison of those multi-choice QA benchmarks, for example,
> > (1) SeViLA achieves higher performance on zero-shot nextqa/star with flant5-3B
> > (2) Videochat2 achieves higher zero-shot performance on star with the similar LLM.
> >
> > Beating absolute sota models generally is not the main research focus/interest from my view, but the effectiveness of the proposed slot design (which is already included in paper ablation studies), however, I believe a comprehensive table (e.g. including connector/LLM types/extra pre-training) provides more interest/solid insights for future module designs.
> >
> > Thanks for the effort again, and I am willing to increase my score based on these rebuttal results.

---

> > > ### Author Response · Authors · 2024-08-08
> > >
> > > Thank you very much for your constructive suggestions. We will include these results in our revised manuscript.
> > >
> > > Moreover, we will follow your insightful suggestions to present more comprehensive comparisons, including comprehensive results and model/training information.
> > >
> > > Thank you again for your great efforts to help us improve!

---

### Author Rebuttal · Authors · 2024-08-02

Dear Reviewers and Area Chair,

Thank you very much for your great efforts and insightful feedback on our paper. **We are grateful for your positive feedback on our motivation/insights (Reviewer jNDE, KJVB), idea novelty and interestingness (Reviewer gzPw, wQQx, KJVB), comprehensive experiments and method effectiveness (Reviewer jNDE, gzPw, wQQx, KJVB), and paper writing and comprehensive details (Reviewer jNDE, gzPw, wQQx).**

We have carefully considered each of your comments and suggestions. We provide detailed responses to address the concerns. Particularly, we summarize our responses regarding each reviewer’s main concerns/comments below:

1)	Based on **Reviewer jNDE**’s concern on ChatGPT-based metrics and suggestion on testing on recent multi-choice QA benchmarks, we have conducted additional zero-shot testing on recent multi-choice QA benchmarks and provide results (see A3). These results demonstrate the robustness and generalizability of our approach.

2)	Based on **Reviewer gzPw**’s suggestions and comments, we have added the ablation study on the influence of frame sampling rate and strides, respectively (see A3). We also show that our method is scalable in handling longer videos, e.g., EgoSchema (see A2). This scalability ensures the applicability of our method to diverse video datasets.

3)	Regarding the questions of **Reviewer wQQx and KJVB** about the event slots, we will clarified the definition and add more explanation in the revision (see A1 to Reviewer wQQx and A1 to Reviewer KJVB). Temporal slots aggregate the temporal tokens for explaining parts of the input, similar to identifying events. Thus, we name the learned slots as event-centric slots. More precisely, we leverage temporal slots to aggregate semantically consistent temporal segments, akin to how spatial slots aggregate spatially consistent regions, termed object-centric slots. While these temporal segments within a slot might not form a complete event, they are likely part of the same event.

4)	Regarding the questions of **Reviewer KJVB**, we have analyzed the reason why Slot-VLM significantly outperforms ChatUniV (see A2) and demonstrated the necessity of the two-branch design (see A4). The analysis underscores the advantages of our architectural choices in enhancing video-language modeling.

We believe that incorporating your valuable insights will significantly enhance the quality and clarity of our paper. We hope our responses adequately address your concerns and look forward to any further feedback you may have. Please feel free to share any additional questions or concerns!


Many thanks,

All authors

---

### Decision · Program_Chairs · 2024-09-25

**Decision:**

Accept (poster)

**Comment:**

The paper received split ratings, 3 weak accepts and 1 borderline reject.

The rebuttal successfully addressed most of the concerns from the 3 reviewers, including additional benchmarks. The borderline reject reviewer's remaining concern is due to the misleading claim of the paper, which the authors promised to fix. The AC does not see a major reason to turn down the majority of reviewers' opinions.

As mentioned during the reviewer discussions, the authors are advised to include the new experimental results in the final version of the paper. Also revise the claim of the event slot in the paper, to more accurately reflect its capabilities and limitations.